# Bayesian Spline Learning for Equation Discovery of Nonlinear Dynamics with Quantified Uncertainty

**Luning Sun**[*]
University of Notre Dame
lsun7@nd.edu

**Daniel Zhengyu Huang**[*]
California Institute of Technology
dzhuang@caltech.edu

**Hao Sun**
Renmin University of China
haosun@ruc.edu.cn

**Jian-Xun Wang** [†]
University of Notre Dame
jwang33@nd.edu

## Abstract

Nonlinear dynamics are ubiquitous in science and engineering applications, but the physics of most complex systems is far from being fully understood. Discovering interpretable governing equations from measurement data can help us understand and predict the behavior of complex dynamic systems. Although extensive work has recently been done in this field, robustly distilling explicit model forms from very sparse data with considerable noise remains intractable. Moreover, quantifying and propagating the uncertainty of the identified system from noisy data is challenging, and relevant literature is still limited. To bridge this gap, we develop a novel Bayesian spline learning framework to identify parsimonious governing equations of nonlinear (spatio)temporal dynamics from sparse, noisy data with quantified uncertainty. The proposed method utilizes spline basis to handle the data scarcity and measurement noise, upon which a group of derivatives can be accurately computed to form a library of candidate model terms. The equation residuals are used to inform the spline learning in a Bayesian manner, where approximate Bayesian uncertainty calibration techniques are employed to approximate posterior distributions of the trainable parameters. To promote the sparsity, an iterative sequential-threshold Bayesian learning approach is developed, using the alternative direction optimization strategy to systematically approximate L0 sparsity constraints. The proposed algorithm is evaluated on multiple nonlinear dynamical systems governed by canonical ordinary and partial differential equations, and the merit/superiority of the proposed method is demonstrated by comparison with state-of-the-art methods.

## 1 Introduction

In the realm of science and engineering, dynamical systems are ubiquitous. However, in actual circumstances, the governing equations behind complicated dynamics may not be completely understood, preventing researchers from developing first-principled models. On the other hand, the ever-increasing data availability opens up new avenues for scientists to identify predictive models from enormous observation data, a process known as *system identification* (SI). Recent advances in deep learning have prompted the rapid development of powerful SI models for high-dimensional problems using deep neural networks (DNNs). Many DNN-based SI models have been proposed to

---

[*]Equal contribution
[†]Corresponding author

36th Conference on Neural Information Processing Systems (NeurIPS 2022).

learn differential operators for complex (spatio)temporal physics from data and shown good potential in terms of data reproduction and state prediction [1–3]. However, deep learning models usually lack interpretability and are difficult to comprehend. Furthermore, it is questionable in terms of generalizability when compared to first-principle models, as such black-box DNN models provide less insight into the underlying processes.

Instead of identifying a black-box model, we focus on extracting analytical equation forms from data, which has higher interpretability and has the potential to advance our knowledge of unknown physics. The sparse identification of non-linear dynamic (SINDy) algorithm [4] is an excellent development along this path. The central idea is to use sparse linear regression to uncover parsimonious governing equations from a dictionary of basis functions constructed by data, where the sparsity is promoted by pruning out redundant terms based on certain specified thresholds [5]. SINDy, although showing great promise, faces several challenges: (1) it heavily relies on high-quality data to extract derivative information, which is typically based on finite difference (FD) methods, making it impossible to handle incomplete, scarce, or noisy data; (2) it is formulated in a deterministic fashion and cannot account for the uncertainties from multiple sources, which is critical for real-world applications where data is frequently corrupted and can be very sparse. In the past few years, the SINDy framework has been further improved in various aspects to address these challenges, e.g., enhancing the library [6] or using deep learning for denoising and derivative computation by fitting the noisy data in a decoupled [7–9] or coupled manner [10–14]. For uncertainty quantification, the dictionary-based equation discovery algorithms have been recently extended to Bayesian settings [15–17], based on the idea of sparse Bayesian learning pioneered by Tipping and co-workers [18–22].

Despite recent progress and extensive work in this field, reliably distilling explicit equation forms from very sparse data with significant noise remains an unsolved challenge. There are still significant gaps in handling data scarcity and noise, quantifying and reducing multi-source uncertainties, and promoting sparsity, which most existing equation discovery techniques struggle to simultaneously address. To this end, we propose a novel Bayesian Spline Learning (BSL) approach to identify parsimonious ordinal/partial differential equations (ODEs/PDEs) from sparse and noisy measurements; meanwhile, the associated uncertainties are quantified. To deal with data scarcity and measurement noise, the proposed BSL uses a spline basis, on which a collection of derivatives can be reliably computed for the library construction. The posterior distributions of spline-based model parameters are approximated by a stochastic gradient descent (SGD) trajectory-based training scheme, where the first moment of SGD iterations is computed by stochastic weight averaging approach [23]. The proposed BSL approach is effective in two aspects. On the one hand, the spline-based representation can help to interpolate locally the spatiotemporal field and perform differentiation analytically. As a result, it considerably enhances learning efficiency in cases with sparse and noisy data. The posterior distributions of spline, library, and equation coefficients are estimated simultaneously, without adding too much overhead to the training process. Besides the measurement uncertainty, the model-form uncertainty is also obtained in the proposed method, which can be used for downstream Bayesian data assimilation, where online data is assimilated to improve the predictability of the identified system for chaotic scenarios.[3]

## 2 Related Work

The data in real-world circumstances is frequently sparse in spatial/temporal domains and may contain considerable noise, posing significant challenges to SINDy or its variants [4–6, 16, 24–27]. Deep learning (DL) has been leveraged as a superb interpolator for concurrently generating metadata and smoothing high-frequency noise [28], effectively improving the performance in identifying equations from imperfect measurements [7, 10–14]. For example, automatic differentiation (AD) is effective for computing derivatives analytically from sparse, noisy data using a point-wise multi-layer perceptron (MLP) [13, 29, 30]. However, it is difficult to impose locality constraints in the point-wise formulation. Wandel et al. [31] demonstrated the merit of employing a spline basis for analytically calculating derivatives while enforcing the locality of spatiotemporal fields. Owing to the superiority of spline-based differentiation compared to numerical discretization, physics-informed spline networks remarkably outperform point-wise physics-informed neural networks (PINN) in the context of solving PDEs [31] and discovering ODEs [14].

---

[3]The code will be available at `https://github.com/luningsun/SplineLearningEquation`

However, these models fail to simultaneously deal with ODE and PDE systems when the data is sparse and substantially corrupted. For the PDE datasets, for example, the PINN-based sparse regression (PINN-SR) [13] works admirably, but it fails to converge on ODE datasets. On the other hand, algorithms such as physics-informed spline learning (PiSL) [14] that performs very well for ODE discovery cannot handle PDE problems due to the limitation of the B-spline basis adopted. More importantly, all these SOTA sparse learning algorithms are formulated in deterministic settings and uncertainties introduced from data/library imperfection cannot be quantified.

For inverse problems like equation discovery, it is natural to use Bayesian framework to quantify and analyze the prediction uncertainty. To enable the posterior computation for high-dimensional trainable parameters that are intractable by traditional Bayesian inference, people have resorted to various approximation strategies, e.g., variational inference [32–34], Monte Carlo dropout [35], Bayes by backprop [36], Laplace approximation [37], and deep ensemble approaches [38]. Although these techniques have had great success, training may still be challenging, and costs will rise dramatically as the problems become more sophisticated. Alternatively, we chose to employ an SGD trajectory-based algorithm, Stochastic Weight Averaging-Gaussian (SWAG) [23], where the information in the SGD trajectory is exploited to approximate the posterior. As demonstrated empirically, SWAC is well scalable to high-dimensional problems and can accurately estimate uncertainty across many different Bayesian learning tasks [23, 39].

Data assimilation (DA) has been widely used in numerical weather prediction (NWP) by fusing online sensing data into a predictive model for nonlinear dynamics forecast. People have recently integrate deep learning into DA to improve online prediction performance [40–42]. However, the predictive model in DA is assumed to be known *a priori*, and uncertainties from the model and observations, which are required for DA, are usually hard to obtain. In our framework, the predictive model can be unknown *a priori* and will be identified to assimilate additional data for online forecasting. Moreover, instead of arbitrarily guessing the observation and model-form errors, this information can be learned in the proposed BSL framework, as the data and model-form uncertainties are quantified. The UQ capability of BSL naturally integrates the equation discovery with Bayesian DA techniques.

The main contributions of this work are three-fold: (1) we extended spline learning for sparse equation discovery of spatiotemporal physics governed by ODEs or PDEs; (2) we developed sparsity-promoting Bayesian learning for UQ tasks; and (3) the proposed BSL framework can seamlessly integrate with Bayesian data assimilation techniques to improve online dynamics forecasting.

## 3  Methodology

Let us consider a dynamical system, which is governed by a parametric ODE/PDE system in the following general form

$$\frac{d\mathbf{u}}{dt} = \mathcal{F}(\mathbf{u}), \tag{1}$$

where $\mathbf{u}$ denotes the state vector, for ODE systems, $\mathbf{u} = [u_1(t), u_2(t), \cdots, u_d(t)]^T \in \mathbb{R}^d$ depends only on time $t$, for PDE systems, $\mathbf{u}(\mathbf{x}, t) = [u_1(\mathbf{x}, t), u_2(\mathbf{x}, t), \cdots, u_d(\mathbf{x}, t)]^d \in \mathbb{R}^d$ depends on both time $t$ and space $\mathbf{x}$, and $\mathcal{F} : \mathbb{R}^d \to \mathbb{R}^d$ represents unknown nonlinear functions. The states are observed at discrete times $\{t_i\}_{i=1}^n$ and at spatial locations $\{\mathbf{x}_j\}_{j=1}^s$ for PDE systems. Let $\tilde{\mathbf{u}}$ denote the noisy observation vector, the observation set is $\tilde{\mathbf{U}} = \{\tilde{\mathbf{u}}(t_1), \tilde{\mathbf{u}}(t_2), \cdots, \tilde{\mathbf{u}}(t_n)\}^T \in \mathbb{R}^{n \times d}$ and $\tilde{\mathbf{U}} = \{\tilde{\mathbf{u}}(\mathbf{x}_1, t_1), \tilde{\mathbf{u}}(\mathbf{x}_2, t_1), \cdots, \tilde{\mathbf{u}}(\mathbf{x}_s, t_n)\}^T \in \mathbb{R}^{(n \times s) \times d}$ for ODE/PDE systems, respectively. Our goal here is to explicitly discover the parsimonious form of $\mathcal{F}(\cdot)$ from a library of candidate basis functions and quantify the associated uncertainty given noisy observation data.

### 3.1  Overview

Given the dataset $\tilde{\mathbf{U}}$, the SI problem can be solved by sparse regression techniques with a predefined library $\boldsymbol{\Phi}(\mathbf{u})$ of $m$ basis functions,

$$\boldsymbol{\Phi}(\mathbf{u}) = [\phi_1(\mathbf{u}), \phi_2(\mathbf{u}), \cdots, \phi_m(\mathbf{u})] \in \mathbb{R}^m, \tag{2}$$

where $\phi_i : \mathbb{R}^d \to \mathbb{R}, 1 \leq i \leq m$ denotes a basis function, which, for instance, can be the polynomial or trigonometric function. Hence, the matrix of library terms evaluated at observed states is defined

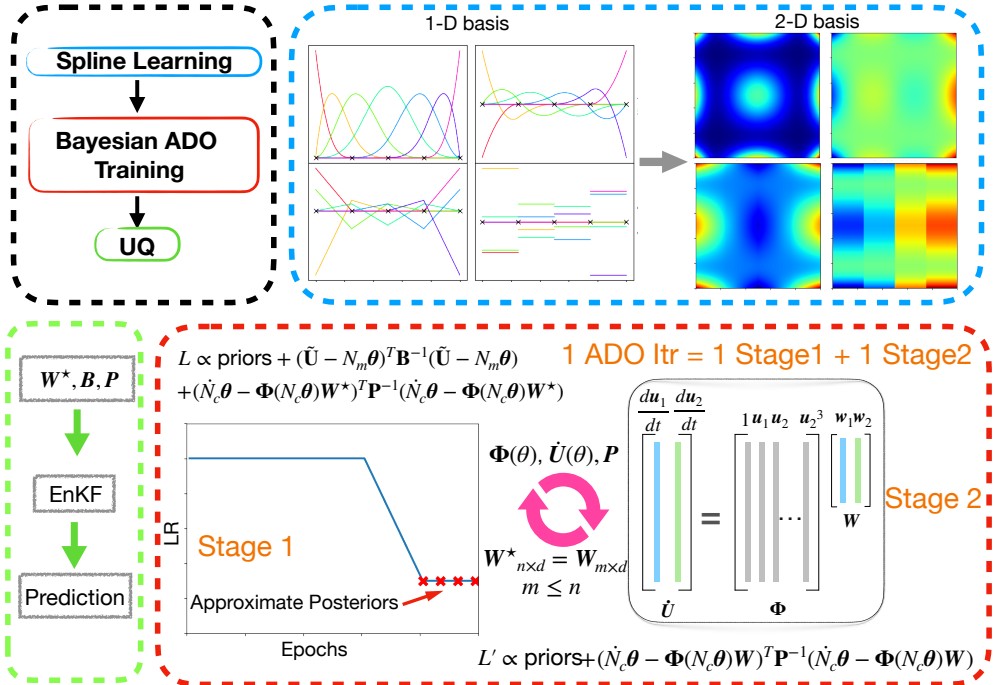

**Figure 1:** Overview of method. Black box: overall work flow. Blue box: a demo for spline basis with derivatives in 1D and 2D cases. Red box: a sketch for the Bayesian ADO training process. Green box: the data assimilation enhanced uncertainty quantification.

as,

$$
\begin{aligned}
\text{ODE}: \quad & \mathbf{\Phi}(\tilde{\mathbf{U}}) = \left[\mathbf{\Phi}\big(\tilde{\mathbf{u}}(t_1)\big)^T, \mathbf{\Phi}\big(\tilde{\mathbf{u}}(t_2)\big)^T, \cdots, \mathbf{\Phi}\big(\tilde{\mathbf{u}}(t_n)\big)^T\right]^T \in \mathbb{R}^{n \times m}, \\
\text{PDE}: \quad & \mathbf{\Phi}(\tilde{\mathbf{U}}) = \left[\mathbf{\Phi}\big(\tilde{\mathbf{u}}(\mathbf{x}_1, t_1)\big)^T, \mathbf{\Phi}\big(\tilde{\mathbf{u}}(\mathbf{x}_2, t_1)\big)^T, \cdots, \mathbf{\Phi}\big(\tilde{\mathbf{u}}(\mathbf{x}_s, t_n)\big)^T\right]^T \in \mathbb{R}^{(n \times s) \times m}.
\end{aligned}
\tag{3}
$$

Given the library, the overview of the BSL framework is shown in Fig. 1. Firstly, a spline-based model is constructed to represent state variables and their derivatives by denoising the observation data, which is illustrated in the blue box and discussed in Sec.3.2. Then the spline-based model and sparse regression are trained simultaneously by alternating direction optimization (ADO), as shown in the red box. Specifically, a single ADO iteration contains two sub-processes, where sub-process I trains the spline-based model using log posterior loss and passes the trainable parameters and the noise estimations to sub-process II, which uses a Bayesian SINDy-like method to prune out redundant terms in the library as defined in Eq. 3, and then updates the number of relevant terms in the training loss for sub-process I. After several ADO iterations, the parsimonious form of the governing equations and the posterior distribution for the coefficients will be estimated. More details about the ADO iterations in the Bayesian framework are shown in Sec. 3.3 and Appendix A.3. Finally, with the estimated posterior, the predictive uncertainty can be quantified by evaluating the identified system with an ensemble of parameters. To further improve the prediction capability, especially for chaotic systems, we propose to leverage data assimilation techniques, which is shown in the green box and discussed in Sec.3.4 and Appendix A.5.

## 3.2 Spline-based learning

Several previous works have already shown the potential of spline-based learning and demonstrated the advantages compared with the classical DL structures, e.g.,(MLP, CNN) [14, 31, 43]. Therefore, we use this structure to smooth the solution fields based on the noisy measurement and then identify the underlying PDE/ODE and calculate the corresponding derivatives from the spline-reconstructed

fields. The B-spline curves, based on the De Boor's algorithm, are defined in a recursive way as:

$$N_{s,0}(t) = \begin{cases} 1, & \text{if } \tau_s \leq t \leq \tau_{s+1}. \\ 0, & \text{otherwise.} \end{cases}$$

$$N_{s,k}(t) = \frac{t - \tau_s}{\tau_{s+k} - \tau_s} N_{s,k-1}(t) + \frac{\tau_{s+k+1} - t}{\tau_{s+k+1} - \tau_{s+1}} N_{s+1,k-1}(t), \tag{4}$$

where $\tau_s$ is the location of knots, $k$ is the degree of polynomial. When $k = 3$, it is the well-used Cubic-B Spline curve. With the defined basis, the spline interpolation can be write as $y(t) = \Sigma_{s=0}^{r+k-1} N_{s,k}(t)\theta_s$. And the number of control points (trainable weights) $\boldsymbol{\theta} \in \mathbb{R}^{r+k}$ is chosen empirically. It can be proved that the derivative of $p$ order B-spline basis is a function of $p-1$ order B-spline, written as:

$$\frac{d}{dt} N_{s,k}(t) = \frac{k}{\tau_{s+k} - \tau_s} N_{s,k-1}(t) - \frac{k}{\tau_{s+k+1} - \tau_{s+1}} N_{s+1,k-1}(t). \tag{5}$$

The proof of Eq. 5 is attached in Appendix A.1. The higher-order derivative can be calculated by recursively using Eq. 5. The analytical derivatives of spline basis are very beneficial for PDE discovery tasks since it always involves constructing library terms containing high order derivatives. With a proper order $p$, the first $p$ derivatives are accurate, and there is no error introduced during the derivation, as opposed to using the numerical methods to approximate derivatives. The spline function and its derivatives are defined in a one-dimensional scenario. It is straightforward to extend to n-dimension by direct using tensor-product. For example, for a two-dimensional problem with spatial-temporal fields, the basis can be defined as

$$N_{s_1,s_2}^{k_1,k_2}(t_i, x_j) = N_{s_1,k_1}(t_i)N_{s_2,k_2}(x_j). \tag{6}$$

Here the two-dimensional basis is denoted by a different style to write it compactly. Similarly, the partial derivative for two-dimensional basis is defined as:

$$\frac{\partial^{(q_1+q_2)} N_{s_1,s_2}^{k_1,k_2}}{\partial t^{(q_1)} \partial x^{(q_2)}}(t_i, x_j) = \frac{d^{(q_1)} N_{s_1,k_1}}{dt^{(q_1)}}(t_i) \frac{d^{(q_2)} N_{s_2,k_2}}{dx^{(q_2)}}(x_j). \tag{7}$$

With the definition of sparse system identification and spline reconstruction, the whole spline learning can be stated as follows: given noisy measurement data $\tilde{\mathbf{U}}$, find the best sets of weights $\boldsymbol{\theta}$ and $\mathbf{W}$ so that data fitting loss and the weakly physics-informed loss can be minimized under sparsity constraints , as shown in Eq. 8:

$$\{\boldsymbol{\theta}, \mathbf{W}\} = \underset{\boldsymbol{\theta}', \mathbf{W}'}{\arg\min} \frac{1}{\mathcal{N}_m} ||\mathbf{N}_m \boldsymbol{\theta}' - \tilde{\mathbf{U}}||_{L_2} + \frac{1}{\mathcal{N}_c} ||\boldsymbol{\Phi}(\mathbf{N}_c \boldsymbol{\theta}')\mathbf{W}' - \dot{\mathbf{N}}_c \boldsymbol{\theta}'||_{L_2} + \lambda ||\mathbf{W}'||_{L_\alpha}. \tag{8}$$

Here, $\mathbf{N}_m$ and $\mathbf{N}_c$ denote the spline basis matrices evaluated at measurement and collocation locations. Moreover, $\mathcal{N}_m$ and $\mathcal{N}_c$ are numbers of measurement data and collocation points. Furthermore, the alternating direction optimization (ADO) shown in previous works [13, 14] can be adopted to minimize the loss function efficiently, and the details are attached in Appendix A.3.

### 3.3 Sparse system identification in Bayesian formulation

Bayesian methods provide a natural probabilistic representation of uncertainty, which is crucial for model predictions. System identification from noisy and sparse measurements generally contains two types of errors (similar to the hidden Markov model): (1) Observation error, where the data is noisy, and the smoothed data is reconstructed through spline-based learning, as shown in Eq. 9

$$\tilde{\mathbf{U}} = \mathbf{N}_m \boldsymbol{\theta}' + \boldsymbol{\epsilon}_1, \tag{9}$$

where $\boldsymbol{\epsilon}_1$ represents the observation error. (2) Evolution error or model form error, since the discovered system cannot be exact due to library imperfection and needs to be reformulated as,

$$\dot{\mathbf{u}} = \boldsymbol{\Phi}(\mathbf{u})\mathbf{W}' + \boldsymbol{\epsilon}_2, \tag{10}$$

where $\boldsymbol{\epsilon}_2$ represents model form error. And Eq. (10) is evaluated on collocation points.

In this work, these error terms are modeled as zero-mean multivariate Gaussian random variables: $\boldsymbol{\epsilon_1} \sim \mathcal{N}(\mathbf{0}, \mathbf{B})$ and $\boldsymbol{\epsilon_2} \sim \mathcal{N}(\mathbf{0}, \mathbf{P})$. And we further assume that these error covariance matrices $\mathbf{B}, \mathbf{P}$

are diagonal matrix with diagonal terms $\{b_k\}$ and $\{p_k\}$ with $1 \leq k \leq d$ and they are learn-able parameters during training.

According to Bayes' rule, the posterior can be written as:

$$p(\boldsymbol{\theta}, \mathbf{W}, \mathbf{B}, \mathbf{P}|\tilde{\mathbf{U}}, \dot{\mathbf{U}}) \propto p(\boldsymbol{\theta}, \mathbf{W}, \mathbf{B}, \mathbf{P})p(\tilde{\mathbf{U}}, \dot{\mathbf{U}}|\mathbf{W}, \boldsymbol{\theta}, \mathbf{B}, \mathbf{P}). \tag{11}$$

Here $\dot{\mathbf{U}} = \dot{\mathbf{N}}_c \boldsymbol{\theta}$ denotes the derivative estimation on all the collocation points. The prior is further decomposed by

$$p(\mathbf{W}, \boldsymbol{\theta}, \mathbf{B}, \mathbf{P}) \propto p(\mathbf{W}|\alpha)p(\alpha)p(\boldsymbol{\theta}|\beta)p(\beta)p(\mathbf{B})p(\mathbf{P}). \tag{12}$$

Currently, we specify the prior for the linear coefficient matrix as a zero mean Gaussian distribution $p(\mathbf{W}|\alpha) = \mathcal{N}(\mathbf{W}|0, \alpha^{-1}\mathbf{I})$ with the hyper prior as a Gamma distribution $p(\alpha) = \text{Gamma}(\alpha|a_0, b_0)$. Similarly, we also define the prior for the spline trainable parameters as zero mean Gaussian distribution with the hyper prior as another Gamma distribution $p(\boldsymbol{\theta}|\beta) = \mathcal{N}(\boldsymbol{\theta}|0, \beta^{-1}\mathbf{I})$ and $p(\beta) = \text{Gamma}(\beta|a_1, b_1)$. To account for the data uncertainty and process uncertainty, the diagonal covariance matrices $\mathbf{B}$ and $\mathbf{P}$ are also set as learn-able during the training. And hence improper uniform priors are used for $\mathbf{B}$ and $\mathbf{P}$.

The likelihood consists of two parts, as:

$$p(\tilde{\mathbf{U}}, \dot{\mathbf{U}}|\mathbf{W}, \boldsymbol{\theta}, \mathbf{B}, \mathbf{P}) = p(\tilde{\mathbf{U}}|\boldsymbol{\theta}, \mathbf{B})p(\dot{\mathbf{U}}|\mathbf{W}, \mathbf{P}, \boldsymbol{\theta}), \tag{13}$$

where

$$p(\tilde{\mathbf{U}}|\boldsymbol{\theta}, \mathbf{B}) \propto \exp\{-\frac{1}{2}(\tilde{\mathbf{U}} - \mathbf{N}_m\boldsymbol{\theta})^T\mathbf{B}^{-1}(\tilde{\mathbf{U}} - \mathbf{N}_m\boldsymbol{\theta})\}, \tag{14}$$

$$p(\dot{\mathbf{U}}|\mathbf{W}, \mathbf{P}, \boldsymbol{\theta}) \propto \exp\{-\frac{1}{2}(\dot{\mathbf{N}}_c\boldsymbol{\theta} - \boldsymbol{\Phi}(\mathbf{N}_c\boldsymbol{\theta})\boldsymbol{W})^T\mathbf{P}^{-1}(\dot{\mathbf{N}}_c\boldsymbol{\theta} - \boldsymbol{\Phi}(\mathbf{N}_c\boldsymbol{\theta})\boldsymbol{W})\}. \tag{15}$$

Traditional Bayesian sampling approaches (i.e., Markov chain Monte Carlo methods) can be usually intractable and expensive, especially when the parameter dimensionality is high. Therefore, researchers tend to use alternative approximation approaches. Current work uses a stochastic gradient descent (SGD) trajectory-based approach, Stochastic Weight Averaging Gaussian (SWAG) [23] algorithm to approximately sample from the posterior distribution. This method approximates the posterior by collecting the parameters near the loss plateau after a sufficient number of the training steps. To further reduce the inference cost, we construct a subspace by finding the PCA components of the SWAG trajectories [44] and then draw samples in the subspace instead. In terms of the loss function in the probabilistic model, we chose to maximize the log form of the posterior density Eq. 11, also by leveraging the ADO algorithm. The sub-process I minimizes the log density function defined by Eq. 11-13 and the sub-process II adopts the Bayesian variants of SINDy algorithm modified from [17, 19]. The likelihood function in sub-process II has the same form as in Eq. 15 but with a different sparsity promoting prior $p(\mathbf{W}|\mathbf{A}) = \prod_{j=1}^{m} \mathcal{N}(\mathbf{W}_j|0, {\alpha'}_j^{-1})$, where $\boldsymbol{A} = [\alpha'_1, \alpha'_2, ..., \alpha'_m]^T$. In a single ADO iteration, the sub-process I provides the updated $\theta$ and $\mathbf{P}$ to sub-process II for constructing the library terms. While the sub-process II shrinks the library terms and passes updated library/weight structure $\mathbf{W}$ back to sub-process I. The whole training requires multiple ADO iterations before it reaches the final balance, where no more terms will be pruned out in sub-process II. The detailed ADO algorithm is listed in Appendix A.3. After obtaining the approximated posterior distribution, predicting uncertainty can be estimated by marginalizing out the model parameters.

### 3.4 Data assimilation for enhanced predictability

For a chaotic system, one notorious problem is that a slight perturb in any model parameters can significantly influence the prediction. For example, in numerical weather prediction (NWP) tasks where the researcher always needs to predict the behavior of chaotic weather systems, various data assimilation (DA) techniques have been developed to assimilate the available data and the known equations. Kalman filter and its variants are successful mathematical tools for data assimilation. Our Bayesian formulation, providing the observation error covariance matrix $\mathbf{B}$ and evolution error covariance matrix $\mathbf{P}$, can be naturally incorporated with the Kalman filter frameworks to improve the predictability of chaotic systems. Current works choose the ensemble Kalman filter for the task, and more background can be found in Appendix A.3.

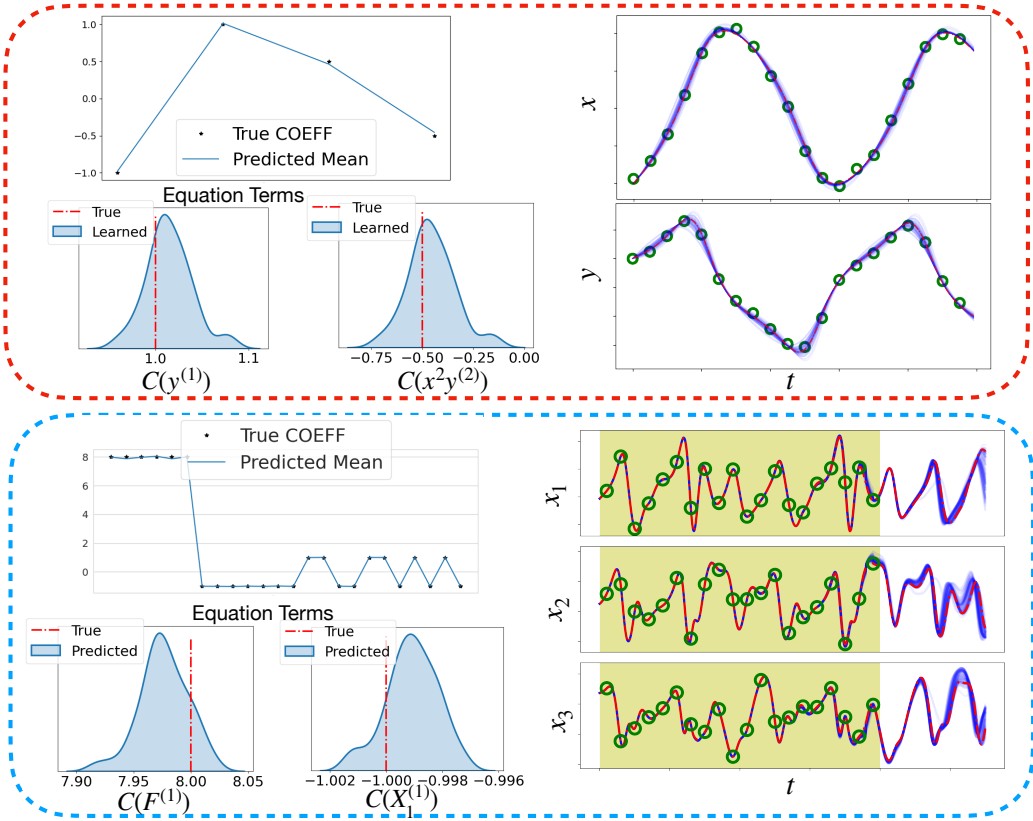

**Figure 2:** The discovery results for ODE systems; the red box shows results for the Van der Pol (VdP) system, and the blue box shows the result for Lorenz 96 system. The layout inside each box follows the rules below. Upper left sub-fig: discovered mean for the relevant library terms; Lower left sub-figs: selected posterior distribution for the identified distributions; Right sub-figs: ensemble prediction plots for UQ. For VdP system, the governing equation is $\frac{dx}{dt} = y$ and $\frac{dy}{dt} = -x - 0.5x^2y + 0.5y$. For the Lorenz96 system, the compact form of the governing equation is $\frac{dX_i}{dt} = (X_{i+1} - X_{i-2})X_{i-1} - X_i + F$ with periodic boundary conditions.

# 4 Experiment and Result

In this section, we first show the equation discovery and uncertainty quantification results for nonlinear ODE systems. We also show that incorporating DA techniques can improve the predicting ability for chaotic systems. Finally, we present the PDE discovery results for several canonical PDE systems.

The first pedagogical ODE example is the Van der Pol oscillator, which is defined as $\frac{dx}{dt} = y$ and $\frac{dy}{dt} = \mu(1 - x^2)y - x$ with $\mu = 0.5$. The data is corrupted with $5\%$ noise, and the library consists of polynomials of state variables up to $3rd$ order. The parsimonious model structure can be correctly identified, and the result is shown in the upper red box of Fig. 2. There are five sub-figures, which depict the library discovery, coefficient distribution, and the forward propagated UQ results. Specifically, the single upper left figure shows the mean of discovered coefficients (blue lines —) and the truth equation coefficients (black stars ∗), where the horizontal axis represents term indices. For example, the Van der Pol system has four different terms, and the x-axis ranges from 1 to 4. The two sub-figures in the lower left part show probability density distributions (PDF) of two identified coefficients, where truth equation coefficients (red lines —) fall within the confidence interval with high probability. The right sub-figures show the propagated ensemble results (blue lines —) based on the discovered equations, measurement data (green circles ○), and the true state trajectories of $x$ and $y$ (red dashed lines — —). The result clearly shows that the ensemble predictions can cover the data, and the uncertainty range fluctuates around the truth state values.

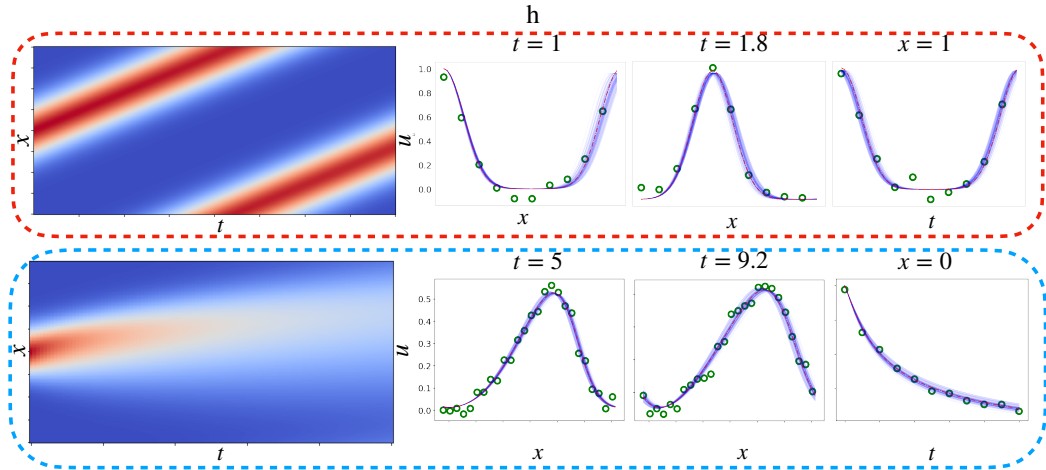

**Figure 3:** The discovery results for PDE systems; the red box shows results for the advection system, and the blue box shows the result for the Burgers' system. The layout inside each box follows the rules below. Leftmost sub-fig: true contour plot; Middle two sub-figs: the spatial results at different time $t$; Rightmost sub-figs: the temporal result at a fixed point $x$.

The second ODE example is a chaotic system, Lorenz 96. It is a simplified mathematical model for atmospheric convection, defined as: $\frac{dX_i}{dt} = (X_{i+1} - X_{i-2})X_{i-1} - X_i + F$, $i = 1, 2, ...n$ with periodic boundary conditions $X_{-1} = X_{n-1}, X_0 = X_n$ and $X_{n+1} = X_1$. We chose $n = 4$ in current case and $F = 8$ for the forcing terms. The measurement states variables are corrupted with $10\%$ Gaussian noise, and the library consists of polynomials of state variables up to $3rd$ order. The discovery result for these systems is shown in the lower blue box in Fig. 2 following the same layout as the Van der Pol systems. The upper left figure indicates that the sparsity structure can be identified, and the horizontal axis marks 24 parsimonious terms (4 terms for each state, 6 states in total) out of 84 library terms. The detailed PDF plots show that posterior distributions still cover truth values with high probability. Although the discovery result is quite accurate, the forward simulations of identified Lorenz 96 system would induce a large phase difference compared with truth trajectories due to the chaotic nature of the underlying system (in Appendix A.5). To alleviate the chaotic behavior and improve the predictability, we seamlessly coupled the ensemble Kalman filter (EnKF), a classical DA technique, to assimilate noisy measurements with the identified system. The observation error matrix $\boldsymbol{B}$ and evolution error matrix $\boldsymbol{P}$ required for the EnKF scheme can be directly passed from the Bayesian spline learning framework. The right subplots show the DA results for the 3 out of 6 state variables to save space. The horizontal axis marks the evolving time, and the vertical axis represents the state variables. In each subplot, the green shaded region indicates the time interval with available noisy measurement data (marked by green circles (○)). And the red solid-line (—) is the true L96 states. Finally, the blue curves (—) are the ensemble predictions. It can be seen that the ensemble predictions inside the region with measurement data almost overlap with the true trajectory. Furthermore, the ensemble uncertainty grows more significantly in the extrapolation region, but the ensemble ranges still fluctuate around the true trajectory. Note that the prediction for the chaotic system with the DA process is much better than directly forward simulating the identified system, where the predictable interval is only about 1 second. More discovery results are attached in Appendix A.5.

Then, we use the proposed BSL framework to identify classical PDE systems and evaluate the performance of two of them in the main text. They include advection equation $\frac{\partial u}{\partial t} + \frac{\partial u}{\partial x} = 0$ and Burgers' equation $\frac{\partial u}{\partial t} + u\frac{\partial u}{\partial x} - 0.5\frac{\partial^2 u}{\partial x^2} = 0$. The qualitative results are shown in Fig. 3 inside the red and blue boxes, respectively. Inside each box, the leftmost contour plot shows the true state value in the spatiotemporal field, and the right three sub-figures show the UQ results for different cross-sections from the left contour plot. The red lines (—) represent the truth value, blue lines (—) are obtained by ensemble predictions, and green circles (○) represent sparse and noisy measurements. The prediction from the identified PDE system is accurate in the spatiotemporal field, and the ensemble fluctuates around the truth state variables.

Finally we benchmark our proposed method with several state-of-the-art discovery algorithms (PINN-SR [13], SINDy [5] and RVM [45]). The error metric is defined as:

$$\mathbf{rmse} = \frac{||\mathbf{C}_{\text{Discovery}} - \mathbf{C}_{\text{True}}||_2}{||\mathbf{C}_{\text{True}}||_2} \tag{16}$$

$$\mathbf{M_P} = \frac{||\mathbf{C}_{\text{Discovery}} \odot \mathbf{C}_{\text{True}}||_0}{||\mathbf{C}_{\text{Discovery}}||_0} \tag{17}$$

$$\mathbf{M_R} = \frac{||\mathbf{C}_{\text{Discovery}} \odot \mathbf{C}_{\text{True}}||_0}{||\mathbf{C}_{\text{True}}||_0} \tag{18}$$

where $\mathbf{C}_{\text{Discovery}}$ are the non-zero mean prediction from the posterior distribution and $\mathbf{C}_{\text{True}}$ are the true coefficients of the governing equations. If the method fails to converge or cannot identify the correct parsimonious form, we will report the final result as **Fail**. The errors are scaled by $\times 10^{-3}$ to have a clear comparison. Two additional metrics, precision $M_P$ and recall $M_R$, are also defined, where the $\odot$ represents element-wise product of vectors and the $l_0$ norm is the non-zero terms in a vector. It can be seen from Table 1 that our BSL method always performs best, when the noise is significant ($> 5\%$), demonstrating its robustness to noise.

The PINN-SR can discover the PDE equation with corrupted data set, but it fails to predict accurate time trajectories for ODE systems. This is also reported in a relevant paper [46], that the plain MLP structure is not satisfactory for predicting time series. The SINDy method can behave much better when large high-quality data exist. However, the SINDy method can easily fail when noise is significant. Our BSL requires fewer parameters and is easier to converge due to the enforcement of locality constraints by the spline basis model. Furthermore, the SINDy and RVM methods require much more data ($100\%$) to train but are still vulnerable to data noise. These benchmark cases show the potential of our BSL for equation discovery tasks. The full table with more benchmark test cases can be found in Tab. 3 in Appendix. We also apply the proposed method on a real-world dataset, as shown in Tab. 9 and Tab. 10 and discuss the effect of smoothing method in Tab. 12 and Tab. 13.

**Table 1:** ODE and PDE discovery comparison

| Name | **rmse**(0%) | **rmse**(1%) | **rmse** (large[4]) | **M_P** | **M_R**[5] | Training Cost[6] |
|------|--------------|--------------|---------------------|---------|------------|------------------|
| | | | Van der Pol Oscillator | | | |
| **BSL(Ours)** | **0.2** | 2.82 | **18.04** | 1 | 1 | $\sim 133(+3)s$ |
| PINN-SR | Fail[7] | Fail | Fail | 0.214 | 0.75 | $\sim 1213s$ |
| SINDy | 1.0 | **1.93** | Fail | 0.267 | 1.0 | $\sim 10s$ |
| RVM | 1.0 | 2.54 | 27.46 | 1 | 1 | $\sim 10s$ |
| | | | Lorenz 96 | | | |
| **BSL(Ours)** | **0.269** | 1.47 | **13.0** | 1 | 1 | $\sim 1654(+438)s$ |
| PINN-SR | Fail | Fail | Fail | 0.5 | 0.22 | $\sim 10788s$ |
| SINDy | 0.4 | 0.64 | Fail | 0.75 | 1 | $\sim 10s$ |
| RVM | 0.4 | **0.6** | 49.7 | 1 | 1 | $\sim 25s$ |
| | | | Advection Equation | | | |
| **BSL(Ours)** | **0.26** | **1** | **1.9** | 1 | 1 | $\sim 946(+233)s$ |
| PINN-SR | 5.9 | 4.5 | 30.4 | 1 | 1 | $\sim 650s$ |
| SINDy | 2.3 | 8.2 | 38.9 | 1 | 1 | $\sim 10s$ |
| RVM | 0.77 | 6.76 | Fail | 0.2 | 1 | $\sim 4s$ |
| | | | Burgers' Equation | | | |
| **BSL(Ours)** | 3.62 | 4.13 | **6.38** | 1 | 1 | $\sim 117(+74)s$ |
| PINN-SR | 10.2 | **3.3** | 10.3 | 1 | 1 | $\sim 512s$ |
| SINDy | 0.826 | Fail | Fail | 1 | 0.5 | $\sim 10s$ |
| RVM | **0.754** | Fail | Fail | 0.1429 | 0.5 | $\sim 4s$ |

## 5 Discussion and Limitation

In this work, we developed a novel Bayesian spline learning (BSL) framework for equation discovery from sparse and noisy datasets with quantified uncertainty. The proposed framework significantly improves SI performance and contributes to the existing literature in the following aspects: Firstly, the use of spline basis enables us to accurately interpolate solution surfaces and analytically compute derivatives to form the candidate library, outperforming other benchmark methods based on finite difference (FD) or auto differentiation (AD), which either suffer from noisy/sparse data or overfitting issues. Secondly, the proposed Bayesian learning formulation notably enhances the robustness for large data noise and sparsity, and meanwhile, quantifies the predictive uncertainty with minimum computational overhead. Moreover, a Bayesian sparsity-promoting ADO iteration strategy is proposed to promote sparsity and recover the parsimonious governing equation as well as the posterior distribution of its coefficients. Last but not least, the Bayesian DA is also integrated into the BSL framework to improve the online predictability of the chaotic systems, which can potentially benefit real-world tasks such as numerical weather forecasting. The proposed framework is evaluated on discovering multiple canonical ODE and PDE systems, and great superiority has been demonstrated in comparison with state-of-the-art methods.

Admittedly, this work still relies on a pre-defined library of candidate terms, and thus the identified system is largely limited to the functional space determined by the user-specified library. In general, for library-based methods, how to design an inclusive but not unnecessarily large library *a priori* is important yet very challenging, which may require prior knowledge of the system to be identified and thus limit their applications for systems involving complex governing physics. Moreover, in this work, only a uniform displacement of knots is used for spline representation, and the tensor product of 1-D splines is adopted for multi-dimensional spline constructions as shown in Tab. 7. These choices are not optimal and have notable limitations for high-dimensional problems. To tackle this issue, we propose to apply more advanced spline learning techniques to reduce the computational cost and improve the scalability. For instance, using deep learning to optimize knot size has shown great promise for lowering computational costs in high-dimensional settings [47, 48], while defining continuous spline kernels and expanding basis in subdomains will allow a significant reduction of trainable parameters [43]. The improvement of spline learning will be explored in our future work. Lastly, similarly to all data-driven models, the proposed method could have a negative societal impact if it is used abusively, particularly for predictive modeling of high-consequence natural systems, where caution should be taken for decision making.

## Acknowledgments and Disclosure of Funding

The authors would like to acknowledge the funds from National Science Foundation, United States of America under award numbers CMMI-1934300 and OAC-2047127, the Air Force Office of Scientific Research (AFOSR), United States of America under award number FA9550-22-1-0065, and startup funds from the College of Engineering at University of Notre Dame in supporting this study

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
