# A  Appendix

## A.1  Proof: analytical form of spline derivatives

**Theorem 1.** *. The derivative of B-spline curve can be analytically evaluated as:*

$$\dot{y}(t) = \Sigma_{s=0}^{r+k-1} \dot{N}_{s,k}(t)\theta_s, \tag{19}$$

*where*

$$\dot{N}_{s,k}(t) = \frac{k}{\tau_{s+k} - \tau_s} N_{s,k-1}(t) - \frac{k}{\tau_{s+k+1} - \tau_{s+1}} N_{s+1,k-1}(t). \tag{20}$$

*Proof.* We will prove by the induction. For the base case $k = 1$, we have

$$\dot{N}_{s,1}(t) = \begin{cases} \frac{1}{\tau_{s+1}-\tau_s} & t \in [\tau_s, \tau_{s+1}] \\ -\frac{1}{\tau_{s+2}-\tau_{s+1}} & t \in [\tau_{s+1}, \tau_{s+2}] \end{cases} = \frac{1}{\tau_{s+1} - \tau_s} N_{s,0}(t) - \frac{1}{\tau_{s+2} - \tau_{s+1}} N_{s+1,0}(t). \tag{21}$$

Let's suppose that the formula holds for $k$ up to $n$. We will prove that this formula also holds for $k = n + 1$. By the definition in Eq. 4 and the chain rule, we can get that:

$$
\begin{aligned}
\dot{N}_{s,n+1}(t) &= \frac{t - \tau_s}{\tau_{s+n+1} - \tau_s} \dot{N}_{s,n}(t) + \frac{N_{s,n}(t)}{\tau_{s+n+1} - \tau_s} + \frac{\tau_{s+n+2} - t}{\tau_{s+n+2} - \tau_{s+1}} \dot{N}_{s+1,n}(t) - \frac{N_{s+1,n}(t)}{\tau_{s+n+2} - \tau_{s+1}} \\
&= \frac{t - \tau_s}{\tau_{s+n+1} - \tau_s} \left( \frac{n}{\tau_{s+n} - \tau_s} N_{s,n-1}(t) - \frac{n}{\tau_{s+n+1} - \tau_{s+1}} N_{s+1,n-1}(t) \right) \\
&\quad + \frac{\tau_{s+n+2} - t}{\tau_{s+n+2} - \tau_{s+1}} \left( \frac{n}{\tau_{s+n+1} - \tau_{s+1}} N_{s+1,n-1}(t) - \frac{n}{\tau_{s+n+2} - \tau_{s+2}} N_{s+2,n-1}(t) \right) \\
&\quad + \frac{1}{\tau_{s+n+1} - \tau_s} N_{s,n}(t) - \frac{1}{\tau_{s+n+2} - \tau_{s+1}} N_{s+1,n}(t) \\
&= \frac{n + 1}{\tau_{s+n+1} - \tau_s} N_{s,n}(t) - \frac{n + 1}{\tau_{s+n+2} - \tau_{s+1}} N_{s+1,n}(t).
\end{aligned}
\tag{22}
$$

$\square$

## A.2  Spline representation

In this section, we give error bounds for spline representation. For simplicity, we consider 1D scenario and assume the target function $u : [0, 1] \to R$ is periodic and defined on the unit interval $\Omega = [0, 1]$. Consider a set of uniform knots $\Gamma : 0 = \tau_0 \le \tau_1 \le \cdots \le \tau_{r+k} = 1$ with . The space of $k$th degree $\{N_{s,k}\}_{s=0}^{r+k-1}$ splines is

$$S^k(\Omega, \Gamma) = \{p | p(t) \text{ is a polynomial of degree } k \text{ in each } (\tau_i, \tau_{i+1})\} \cap \mathcal{C}^{k-1}(\Omega).$$

Spline interpolation seeks $\hat{u} \in S^k(\Omega, \Gamma)$ that satisfies $u(\tau_i) = \hat{u}(\tau_i) \quad \forall \quad 0 \le r + k$.

**Theorem 2** (Spline interpolation error bounds [49, 50])**.** *Assume that $u$ is periodic, for odd degree spline interpolation $k$, we have*

$$
\begin{aligned}
\|u - \hat{u}\|_\infty &= \Delta\tau^{k+1} \left( C_{k+1} \|u^{(k+1)}\|_\infty + \mathcal{O}(\omega(u^{(k+1)}, \Delta\tau)) \right), \\
\|u^{(l)} - \hat{u}^{(l)}\|_\infty &= \Delta\tau^{k+1-l} \left( D_{k+1-l} \|u^{(k+1)}\|_\infty + \mathcal{O}(\omega(u^{(k+1)}, \Delta\tau)) \right) \quad \forall 1 \le l \le k - 1.
\end{aligned}
\tag{23}
$$

*Here $C_{k+1}$ and $D_{k+1-l}$ are constant parameters, which are independent of $\Delta\tau$ and $u$, and $\omega(u^{(k+1)}, \Delta\tau) = \sup_{|x-y| \le \Delta\tau} |u^{(k+1)}(x) - u^{(k+1)}(y)|$.*

In the present work, we focus on using spline for smoothing noisy data. It is essentially a nonparameteric estimation of function $u$ from noisy data $\{t_i, \tilde{u}_i\}_{i=1}^n$,

$$\tilde{u}_i = u(t_i) + \eta_i \quad \text{with} \quad \eta_i \sim \mathcal{N}(0, \delta_\eta^2).$$

We further assume that $\{t_i\}$ are uniformly sampled from $\Omega$. The unregularized smoothing process estimates spline basis coefficients

$$\hat{\boldsymbol{\theta}} = (\mathbf{N}_m^T \mathbf{N}_m)^{-1} \mathbf{N}_m^T \cdot \tilde{\mathbf{U}}$$

through minimizing $\|\tilde{\mathbf{U}} - \mathbf{N}_m \cdot \boldsymbol{\theta}\|$. Here $\tilde{\mathbf{U}} = \left[\tilde{u}_1, \tilde{u}_2, \cdots, \tilde{u}_n\right]^T$ and $\mathbf{N}_m$ is the spline basis matrix evaluated at these measurement locations. The optimal $\boldsymbol{\theta}^{\mathrm{opt}}$ satisfies $\hat{u}(t) = \mathbf{N}(t) \cdot \boldsymbol{\theta}^{\mathrm{opt}}$, where $\hat{u}$ is the spline interpolation of $u$, and $\mathbf{N}(t) = [N_{0,k}(t), N_{1,k}(t), \cdots, N_{r+k-1,k}(t)]^T$ denotes spline basis function vector. Following [51], we have spline fitting error bounds, as following.

**Theorem 3** (Spline fitting error bounds). *Assume that $u$ is periodic and the number of data $n$ is sufficient large, for odd degree spline interpolation $k$, we have*

$$\|\mathbb{E}\hat{\boldsymbol{\theta}} - \boldsymbol{\theta}^{\mathrm{opt}}\|_2 \leq C_1 \Delta \tau^{k+1} \qquad \|\mathrm{Cov}\hat{\boldsymbol{\theta}}\|_2 \leq C_2 \frac{\delta_\eta^2}{n},$$

*where $C_1$ and $C_2$ are constant and independent of $n$.*

*Proof.* Let us denote

$$\mathbf{U} = [u(t_1), u(t_2), \cdots, u(t_n)]^T \quad \hat{\mathbf{U}} = [\hat{u}(t_1), \hat{u}(t_2), \cdots, \hat{u}(t_n)]^T$$

$$\boldsymbol{\eta} = [\eta_1, \eta_2, \cdots, \eta_n]^T \quad \mathbf{e} = \mathbf{U} - \hat{\mathbf{U}}.$$

We have

$$\begin{aligned}
\hat{\boldsymbol{\theta}} - \boldsymbol{\theta}^{opt} &= (\mathbf{N}_m^T \mathbf{N}_m)^{-1} \mathbf{N}_m^T \cdot \tilde{\mathbf{U}} - (\mathbf{N}_m^T \mathbf{N}_m)^{-1} \mathbf{N}_m^T \mathbf{N}_m \boldsymbol{\theta}^{\mathrm{opt}} \\
&= (\mathbf{N}_m^T \mathbf{N}_m)^{-1} \mathbf{N}_m^T \cdot (\mathbf{U} + \boldsymbol{\eta} - \hat{\mathbf{U}}) \qquad (24)\\
&= (\mathbf{N}_m^T \mathbf{N}_m)^{-1} \mathbf{N}_m^T \cdot (\mathbf{e} + \boldsymbol{\eta}).
\end{aligned}$$

We will first prove that

$$\|\mathbf{N}_m^T \mathbf{N}_m\|_2 = \mathcal{O}(n) \qquad \|(\mathbf{N}_m^T \mathbf{N}_m)^{-1}\|_2 = \mathcal{O}(\frac{1}{n}). \qquad (25)$$

For any $\boldsymbol{\theta}$, $\frac{1}{n} \boldsymbol{\theta}^T \mathbf{N}_m^T \mathbf{N}_m \boldsymbol{\theta}$ is the Monte Carlo approximation of $\int \left(\mathbf{N}(t)^T \cdot \boldsymbol{\theta}\right)^2 dt$, and hence

$$\frac{1}{n} \boldsymbol{\theta}^T \mathbf{N}_m^T \mathbf{N}_m \boldsymbol{\theta} = \int (\mathbf{N}(t)^T \boldsymbol{\theta})^2 dt + \mathcal{O}(\frac{1}{\sqrt{n}}). \qquad (26)$$

Bringing the following property of B-splines [52]

$$M_1 \boldsymbol{\theta}^T \boldsymbol{\theta} \leq \int (\mathbf{N}(t)^T \boldsymbol{\theta})^2 dt \leq M_2 \boldsymbol{\theta}^T \boldsymbol{\theta} \qquad \exists M_1, M_2 > 0$$

into Eq. (26) leads to Eq. (25). Then we prove that

$$\|\mathbf{N}_m^T \mathbf{e}\|_2 = \mathcal{O}(n\Delta \tau^{k+1}). \qquad (27)$$

Since $\sum_s N_{s,k}(t) = 1$ and $\|\mathbf{e}\|_\infty = \mathcal{O}(\Delta \tau^{k+1})$, we have

$$\|\mathbf{N}_m^T \mathbf{e}\|_2 \leq \|\mathbf{N}_m^T \mathbf{e}\|_1 = \mathcal{O}(n\Delta \tau^{k+1}).$$

Finally, combining Eq. (24), Eq. (25) and Eq. (27) leads to

$$\|\mathbb{E}\hat{\boldsymbol{\theta}} - \boldsymbol{\theta}^{\mathrm{opt}}\|_2 = \|(\mathbf{N}_m^T \mathbf{N}_m)^{-1} \mathbf{N}_m^T \cdot \mathbf{e}\|_2 \leq \|(\mathbf{N}_m^T \mathbf{N}_m)^{-1}\|_2 \|\mathbf{N}_m^T \cdot \mathbf{e}\|_2 \leq C_1 \Delta \tau^{k+1}$$

$$\|\mathrm{Cov}\hat{\boldsymbol{\theta}}\|_2 = \sigma_\eta^2 \|(\mathbf{N}_m^T \mathbf{N}_m)^{-1}\|_2 \leq C_2 \frac{\delta_\eta^2}{n}$$

$\square$

In our sparse Bayesian regression, in stead of solving the aforementioned minimization problem, we have additional regularization terms.

## A.3 Algorithms

In this section, we present detailed algorithms used in the present work, which include Bayesian Alternative Direction Optimization (ADO) Learning 1, Sequential Threshold Sparse Bayesian Learning 2, and Ensemble Kalman Filter 3.

---

**Algorithm 1:** Bayesian Alternative Direction Optimization (ADO) Learning

---

**Input** : Library $\boldsymbol{\Phi}$, spline basis $\mathbf{N}$, time derivative of spline basis matrix $\dot{\mathbf{N}}_c$, negative log form of equation Eq. 11 $\mathcal{L}$

**Output**: Mean estimation: $\boldsymbol{\theta}_{\text{SWA}}, \mathbf{W}_{\text{SWA}}, \mathbf{B}_{\text{SWA}}, \mathbf{P}_{\text{SWA}}$

Samples from posterior distributions: $\mathring{\boldsymbol{\theta}}, \mathring{\mathbf{W}}, \mathring{\mathbf{B}}, \mathring{\mathbf{P}}$

**Pretrain**:

**for** $i = 1 : T_{\text{Pretrain}}$ **do**

    **SDG optimization with the fixed library**

    $\{\boldsymbol{\theta}_{i+1}, \mathbf{W}'_{i+1}, \mathbf{B}_{i+1}, \mathbf{P}'_{i+1}\} = \arg \min \mathcal{L}$

    **Update library**

    $\mathbf{W}_{i+1}, \mathbf{P}_{i+1} = \text{STSparseBayesian}(\boldsymbol{\Phi}, \dot{\mathbf{U}}_{i+1} = \dot{\mathbf{N}}_c \boldsymbol{\theta}_{i+1}, \mathbf{W}'_{i+1}, \mathbf{P}'_{i+1})$

    **if** $\mathcal{L}(\boldsymbol{\theta}_{i+1}, \mathbf{W}_{i+1}, \mathbf{B}_{i+1}, \mathbf{P}_{i+1}) < \mathcal{L}^\star$ **then**

        $\mathcal{L}^\star = \mathcal{L}(\boldsymbol{\theta}_{i+1}, \mathbf{W}_{i+1}, \mathbf{B}_{i+1}, \mathbf{P}_{i+1})$

        $\boldsymbol{\theta}^\star, \mathbf{W}^\star, \mathbf{B}^\star, \mathbf{P}^\star = \boldsymbol{\theta}_{i+1}, \mathbf{W}_{i+1}, \mathbf{B}_{i+1}, \mathbf{P}_{i+1}$

    **else**

        break

**end**

**Stochastic Weight Averaging-Gaussian (SWAG) for posterior approximation**:

$\boldsymbol{\theta}_{\text{SWA}}, \mathbf{W}_{\text{SWA}}, \mathbf{B}_{\text{SWA}}, \mathbf{P}_{\text{SWA}} = \boldsymbol{\theta}^\star, \mathbf{W}^\star, \mathbf{B}^\star, \mathbf{P}^\star$

With a constant learning rate **for** $i = 1 : T_{\text{SWAG}}$ **do**

    SGD update $\boldsymbol{\theta}_i, \mathbf{W}_i, \mathbf{B}_i, \mathbf{P}_i$

    $\boldsymbol{\theta}_{\text{SWA}}, \mathbf{W}_{\text{SWA}}, \mathbf{B}_{\text{SWA}}, \mathbf{P}_{\text{SWA}} = \frac{i\boldsymbol{\theta}_{\text{SWA}} + \boldsymbol{\theta}_i}{i+1}, \frac{i\mathbf{W}_{\text{SWA}} + \mathbf{W}_i}{i+1}, \frac{i\mathbf{B}_{\text{SWA}} + \mathbf{B}_i}{i+1}, \frac{i\mathbf{P}_{\text{SWA}} + \mathbf{P}_i}{i+1}$

**end**

**Compute low-rank square root of empirical covariance matrices** $\boldsymbol{\Lambda}_\theta, \boldsymbol{\Lambda}_W, \boldsymbol{\Lambda}_B, \boldsymbol{\Lambda}_P$ from

$\{\boldsymbol{\theta}_i - \boldsymbol{\theta}_{\text{SWA}}\}, \{\boldsymbol{W}_i - \boldsymbol{W}_{\text{SWA}}\}, \{\boldsymbol{B}_i - \boldsymbol{B}_{\text{SWA}}\}, \{\boldsymbol{P}_i - \boldsymbol{P}_{\text{SWA}}\}$

**Sampling**:

$\mathring{\boldsymbol{\theta}} = \boldsymbol{\theta}_{\text{SWA}} + \boldsymbol{\Lambda}_\theta \mathring{z}_\theta \qquad \mathring{\boldsymbol{W}} = \boldsymbol{W}_{\text{SWA}} + \boldsymbol{\Lambda}_W \mathring{z}_W \qquad \mathring{\boldsymbol{B}} = \boldsymbol{B}_{\text{SWA}} + \boldsymbol{\Lambda}_B \mathring{z}_B \qquad \mathring{\boldsymbol{P}} = \boldsymbol{P}_{\text{SWA}} + \boldsymbol{\Lambda}_P \mathring{z}_P$

where $\mathring{z}_*$ are the random samples from $\mathcal{N}(0, I)$.

---

---

**Algorithm 2:** Sequential Threshold Sparse Bayesian Learning

---

**Input** : Spline trainable parameter $\boldsymbol{\theta}$, library $\boldsymbol{\Phi}(\boldsymbol{\theta})$, approximated derivative $\dot{\mathbf{U}}$, library weight $\mathbf{W}$, and process error matrix $\mathbf{P}$

**Output**: Best solution library $\boldsymbol{\Phi}^\star, \mathbf{W}^\star, \mathbf{P}^\star$

**Initialize**: Threshold $\epsilon$, number of library terms $p_{\text{old}}$, and Flag = **True**

**while** *Flag is True* **do**

    **while** *not converged* **do**

        1. Compute the relevance variable $\eta_i = q_i{}^2 - s_i$ as defined in [19]

        2. Update library $\boldsymbol{\Phi}^\star$, weight $\mathbf{W}^\star$, and process error matrix $\mathbf{P}^\star$, as shown in [19]

    **end**

    **for** $j = 1 : p_{old}$ **do**

        $\mathbf{W}^\star(j) = 0$ **If** $|\mathbf{W}^\star(j)| \leq \epsilon$

    **end**

    Find the nonzero entries in $\mathbf{W}^\star$, record the index as $\mathbf{I}$, update $\boldsymbol{\Phi} = \boldsymbol{\Phi}(:, \mathbf{I})$ and $p_{\text{new}}$ = length of $\mathbf{I}$;

    **if** $\mathbf{p_{new}} = \mathbf{p_{old}}$ **then**

        Flag = **False**

**end**

---

**Algorithm 3:** Ensemble Kalman Filter

---

**Input** : ensemble number $J$, sampled weights $\{\mathbf{W}^j\}_{j=1}^J$, discovered dynamical model $\mathbf{M}(\cdot\,;\,\mathbf{W})$,
process noise covariance $\mathbf{P}$, observation model $\mathbf{h}$, observation $\tilde{\mathbf{U}}$, observation noise covariance $\mathbf{B}$

**Output** : Analysis ensemble trajectories $\mathbf{U}_a^j(t)$

**Forecast** : $\mathbf{U}_f^j(t_{i+1}) = \mathbf{M}(\mathbf{U}_a^j(t_i); \mathbf{W}^j) + \boldsymbol{\epsilon}_2^j, \quad \boldsymbol{\epsilon}_2^j \sim (\mathbf{0}, \mathbf{P})$

$\overline{\mathbf{U}}_f(t_{i+1}) = \frac{1}{J}\sum_{j=1}^J \mathbf{U}_f^j(t_{i+1})$

**Analysis** : $\mathbf{U}_h^j(t_{i+1}) = \mathbf{h}(\mathbf{U}_f^j(t_{i+1})) \qquad \overline{\mathbf{U}}_h(t_{i+1}) = \frac{1}{J}\sum_{j=1}^J \mathbf{U}_h^j(t_{i+1})$

$\mathbf{C}^{fh}(t_{i+1}) = \frac{1}{J-1}\Sigma_{j=1}^J(\mathbf{U}_f^i(t_{i+1}) - \overline{\mathbf{U}}_f(t_{i+1}))(\mathbf{U}_h^i(t_{i+1}) - \overline{\mathbf{U}}_h(t_{i+1}))^T$

$\mathbf{C}^{hh}(t_{i+1}) = \frac{1}{J-1}\Sigma_{j=1}^J(\mathbf{U}_h^i(t_{i+1}) - \overline{\mathbf{U}}_h(t_{i+1}))(\mathbf{U}_h^i(t_{i+1}) - \overline{\mathbf{U}}_h(t_{i+1}))^T + \mathbf{B}$

$\mathbf{K}(t_{i+1}) = \mathbf{C}^{fh}(\mathbf{C}^{hh})^{-1}$

$\mathbf{U}_a^j(t_{i+1}) = \mathbf{U}_f^j(t_{i+1}) + \mathbf{K}(t_{i+1})\Big(\tilde{\mathbf{U}}(t_{i+1}) - \mathbf{U}_h^j(t_{i+1}) - \boldsymbol{\epsilon}_1^j\Big), \quad \boldsymbol{\epsilon}_1^j \sim (\mathbf{0}, \mathbf{B})$

---

## A.4 Training Details

Additional training hyper parameters used in Sec. 4 is shown in the Tab. 2.

**Table 2:** Training Details

| Case | Van der Pol | Lorenz 96 | Advection | Burgers' |
|------|-------------|-----------|-----------|----------|
| ADO Iter | 5 | 5 | 1 | 5 |
| ADO Epoch | $20K$ | $50K$ | $20K$ | $20K$ |
| Post Epoch | $1K$ | $65K$ | $2K$ | $0.5K$ |
| SWAG Epoch | $1.5K$ | $80K$ | $0.5K$ | $0.5K$ |
| LR | $1 \times 10^{-2}$ | $1 \times 10^{-2}$ | $1 \times 10^{-2}$ | $1 \times 10^{-2}$ |
| SWAG LR | $1 \times 10^{-3}$ | $1 \times 10^{-3}$ | $1 \times 10^{-3}$ | $1 \times 10^{-3}$ |

## A.5 Additional Result: ODE

We list additional discovery and UQ results in this section. Fig. 5 shows 4 distributions of the coefficients for Van der Pol system in red box and Lorenz 96 system in blue box. Fig. 4 shows additional UQ result from the identified L96 systems without incorporating the data assimilation process. The truth trajectory is marked by red. The measurement is marked by green dots and the ensemble trajectories are marked by blue. Although the system has been identified with high accuracy, as shown in Tab. 3, the predicted ensembles of the state variables still become chaotic after several seconds. It is inevitable since the chaotic nature of the underlying system, which means any small perturbation in any parameters would significantly influence the future trajectories. Fortunately, the predicted covariance matrix of the Bayesian framework makes it easy to incorporate the data assimilation with the identified systems. With the identified distribution of system coefficients, the data assimilation can be used to predict the future states with reduced uncertainty, given noisy measurement data in the past. Fig. 6 shows additional UQ result for all the 6 state variables for Lorenz 96 system, incorporating EnKF algorithms.

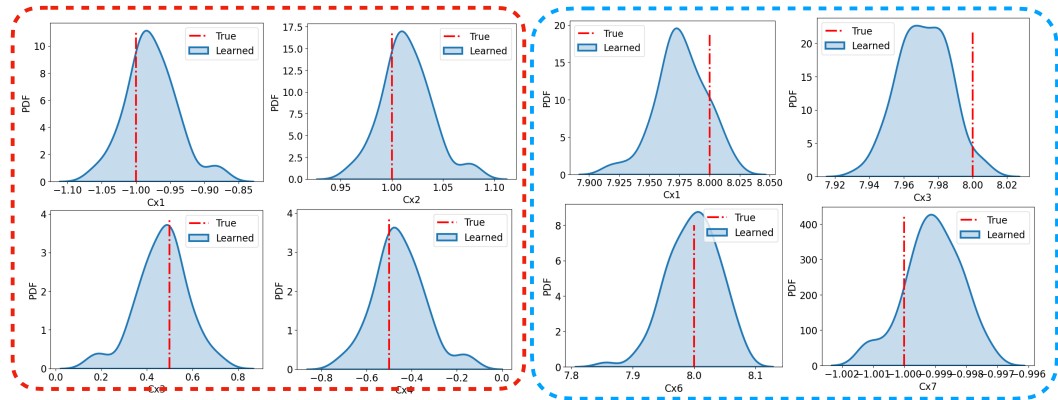

**Figure 4:** Additional discovery results for ODE systems; the red box shows results coefficients distribution for the Van der Pol system, and the blue box shows the result for Lorenz 96 system.

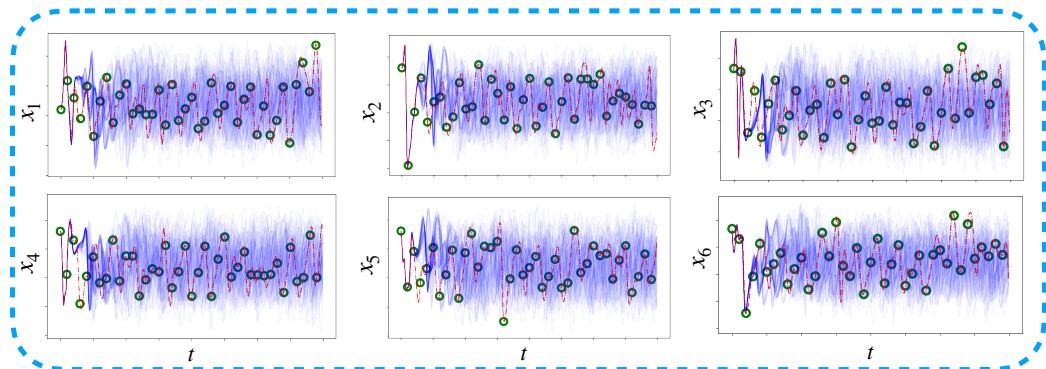

**Figure 5:** Additional UQ results for ODE systems; the blue box shows the all the states prediction for Lorenz 96 system without ensemble Kalman filter.

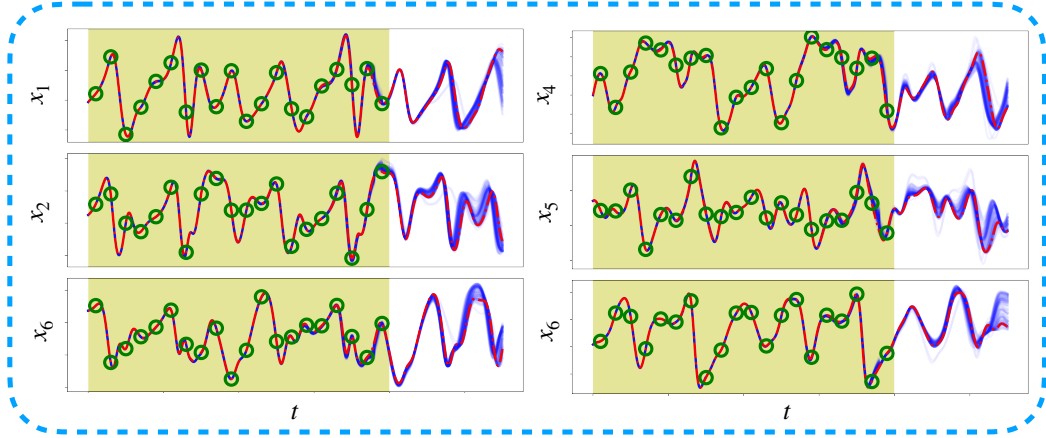

**Figure 6:** Additional UQ results for ODE systems; the blue box shows the all the states prediction for Lorenz 96 system with ensemble Kalman filter.

## A.6   Additional Result: PDE

In this section, we attached the qualitative result for PDE discovery and the uncertainty quantification. The contour plot and the cross section result are shown in Fig 7 (for advection equation and Burgers

equation) and Fig. 8 (for Burgers equation with source). The analytical form of the mentioned PDEs are listed in Tab. 5. Probability distribution for the PDE coefficient are shown in Fig. 9. Additional UQ prediction result for advection and Burgers' equation are shown in and Fig. 10.

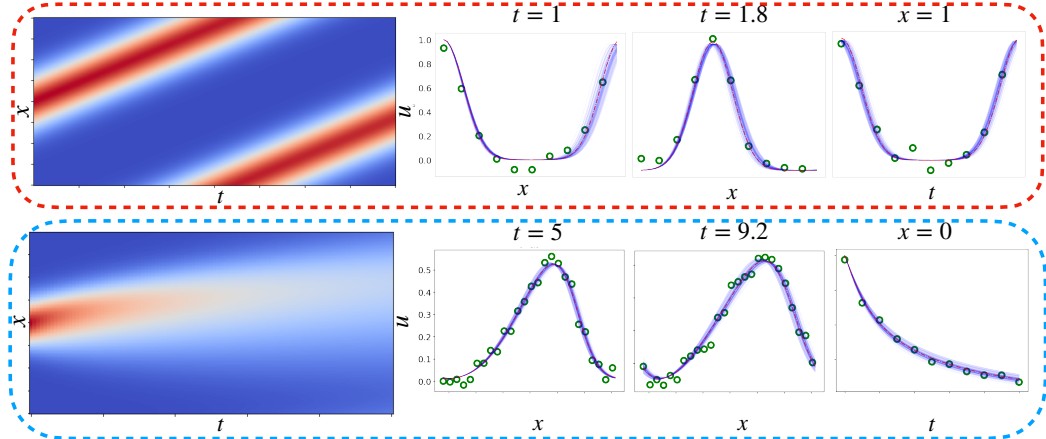

**Figure 7:** The discovery results for PDE systems; the red box shows results for the advection system, and the blue box shows the result for the Burgers' system. The layout inside each box follows the rules below. Leftmost sub-fig: true contour plot; Middle two sub-figs: the spatial results at different time $t$; Rightmost sub-figs: the temporal result at a fixed point $x$.

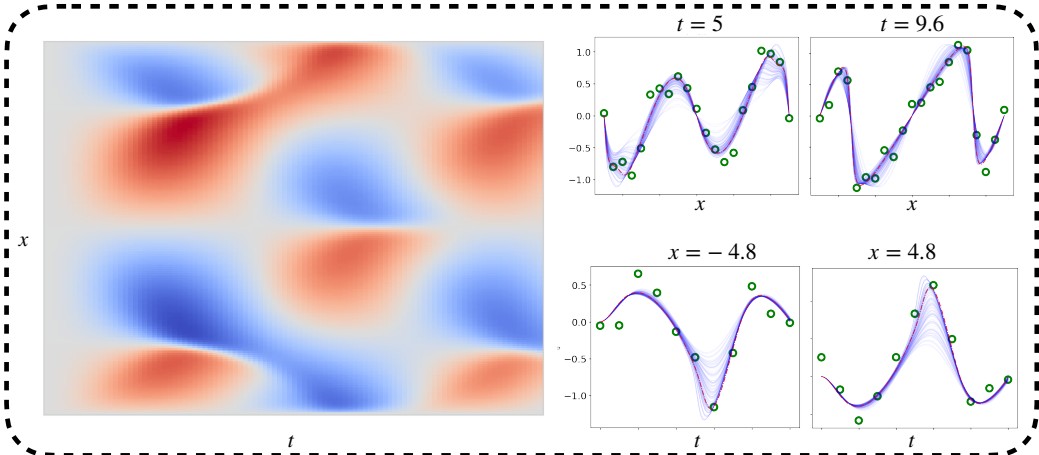

**Figure 8:** Additional UQ results for PDE; the black box shows the cross section UQ results for Burgers' equation with source.

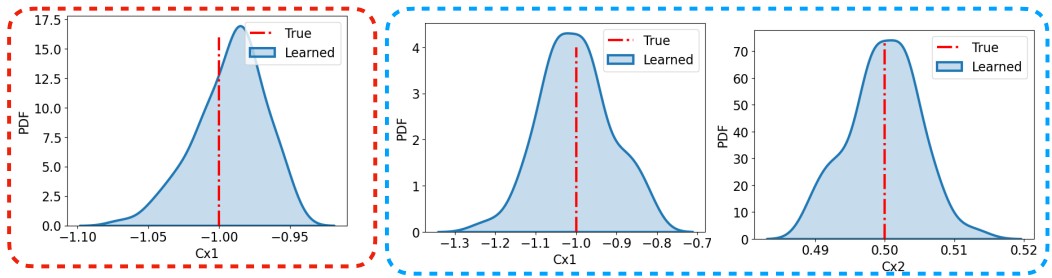

**Figure 9:** Additional discovery results for PDE systems; the red box shows results coefficients distribution for the advection equation, and the blue box shows the result for Burgers' equation.

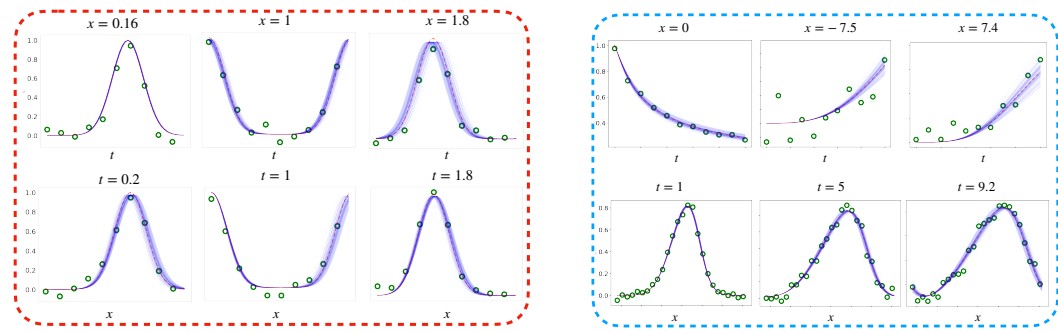

**Figure 10:** Additional UQ results PDE; the red box shows the cross section UQ results for Advection equation. The blue box shows the cross section UQ results for Burgers' equation.

## A.7 Additional Discovery result

In this section, we list the full table that includes all the experiments made for current work, as attached in Tab. 3.

**Table 3:** ODE and PDE discovery comparison

| Name | **rmse**$(0\%)$ | **rmse**$(1\%)$ | **rmse** (large[8]) | $\mathbf{M_P}$ | $\mathbf{M_R}$[9] | Training Cost[10] |
|---|---|---|---|---|---|---|
| | | Van der Pol Oscillator | | | | |
| **BSL(Ours)** | **0.2** | 2.82 | **18.04** | **1** | **1** | $\sim 133(+3)s$ |
| PINN-SR | Fail[11] | Fail | Fail | 0.214 | 0.75 | $\sim 1213s$ |
| SINDy | 1.0 | **1.93** | Fail | 0.267 | 1.0 | $\sim 10s$ |
| RVM | 1.0 | 2.54 | 27.46 | **1** | **1** | $\sim 10s$ |
| | | Lorenz 96 | | | | |
| **BSL(Ours)** | **0.269** | 1.47 | **13.0** | **1** | **1** | $\sim 1654(+438)s$ |
| PINN-SR | Fail | Fail | Fail | 0.5 | 0.22 | $\sim 10788s$ |
| SINDy | 0.4 | 0.64 | Fail | 0.75 | 1 | $\sim 10s$ |
| RVM | 0.4 | **0.6** | 49.7 | **1** | **1** | $\sim 25s$ |
| | | Advection Equation | | | | |
| **BSL(Ours)** | **0.26** | **1** | **1.9** | **1** | **1** | $\sim 946(+233)s$ |
| PINN-SR | 5.9 | 4.5 | 30.4 | **1** | **1** | $\sim 650s$ |
| SINDy | 2.3 | 8.2 | 38.9 | **1** | **1** | $\sim 10s$ |
| RVM | 0.77 | 6.76 | Fail | 0.2 | **1** | $\sim 4s$ |
| | | Burgers' Equation | | | | |
| **BSL(Ours)** | 3.62 | 4.13 | **6.38** | **1** | **1** | $\sim 117(+74)s$ |
| PINN-SR | 10.2 | **3.3** | 10.3 | **1** | **1** | $\sim 512s$ |
| SINDy | 0.826 | Fail | Fail | **1** | 0.5 | $\sim 10s$ |
| RVM | **0.754** | Fail | Fail | 0.1429 | 0.5 | $\sim 4s$ |
| Name | **rmse**$(0\%)$ | **rmse**$(0.1\%)$ | **rmse** (large) | $\mathbf{M_P}$ | $\mathbf{M_R}$ | Training Cost |
| | | Burgers' with Source | | | | |
| **BSL(Ours)** | 11 | **12.4** | **13.4** | **1** | **1** | $\sim 396(+340)s$ |
| PINN-SR | **10.5** | 15 | 34.6 | 1 | 1 | $\sim 600s$ |
| SINDy | 26.2 | Fail | Fail | **1** | 0.33 | $\sim 10s$ |
| RVM | 27.6 | Fail | Fail | 0.5 | 0.67 | $\sim 10s$ |
| | | Heat Equation | | | | |
| **BSL(Ours)** | 19 | 19 | **38.9** | **1** | **1** | $\sim 71(+8)s$ |
| PINN-SR | Fail | Fail | Fail | 0 | 0 | $\sim 285s$ |
| SINDy | **1.9** | **17** | Fail | 0.25 | 1 | $\sim 10s$ |
| RVM | 2.8 | 6 | Fail | 0 | 0 | $\sim 6s$ |
| | | Poisson Equation | | | | |
| **BSL(Ours)** | $\mathbf{1.18 \times 10^{-2}}$ | **0.133** | **16.7** | **1** | **1** | $\sim 92(+9)s$ |
| PINN-SR | Fail | Fail | Fail | 0 | 0 | $\sim 3737s$ |
| SINDy | 1.15 | 87 | 962 | **1** | **1** | $\sim 10s$ |
| RVM | 1.15 | 232 | 968 | **1** | **1** | $\sim 10s$ |

---

[8]Large noise for different cases: Van der Pol: 5%, Lorenz 96: 10%, Advection: 20%, Burgers: 10%, Burgers' with source: 20%, Heat: 15%, Poisson: 5%

[9]$\mathbf{M_P}, \mathbf{M_R}$ are only reported for the largest noise cases

[10]All cases are running on a Nvidia 2070 Ti GPU card

[11]Fail means failure in discovery of the parsimonious ODE/PDE forms.

## A.8 Analytical forms of the discovered system

**Table 4:** Analytical forms of ODE

| Name | $\epsilon$ (large) |
|---|---|
| | Vander Pol Oscillator |
| True | $\frac{dx}{dt} = y, \frac{dy}{dt} = -x - 0.5x^2y + 0.5y$ |
| BSL(Ours) | $\frac{dx}{dt} = 1.0096(\pm0.024)y, \frac{dy}{dt} = -0.9858(\pm0.037)x - 0.4801(\pm0.114)x^2y + 0.4889(\pm0.111)y$ |
| PINN-SR | $\frac{dx}{dt} = 1.3079 - 0.1151x + 0.9982y + 0.1939x^2 - 0.4101xy + 0.8559y^2 - 0.3577x^3$ $+0.6764x^2y - 0.1207xy^2 + 0.6261y^3,$ $\frac{dy}{dt} = -0.6718x + 1.6035y + 1.6339y^2 + 0.4803y^3$ |
| SINDy | $\frac{dx}{dt} = 0.1197 + 0.169x + 0.9975y - 0.0292x^2 + 0.0232xy - 0.0274y^2$ $-0.0393x^3 + 0.0208x^2y - 0.0463xy^2,$ $\frac{dy}{dt} = -1.0909x + 0.149y + 0.0294x^3 - 0.4x^2y - 0.0356xy^2 + 0.0762y^3$ |
| RVM | $\frac{dx}{dt} = 0.9957(\pm0.073)y, \frac{dy}{dt} = -0.9943(\pm0.0597)x - 0.4777(\pm0.0809)x^2y + 0.4714(\pm0.1204)y$ |
| | Lorenz 96 |
| True | $\frac{dX_1}{dt} = (X_2 - X_5)X_6 - X_1 + 8, \frac{dX_2}{dt} = (X_3 - X_6)X_1 - X_2 + 8,$ $\frac{dX_3}{dt} = (X_4 - X_1)X_2 - X_3 + 8, \frac{dX_4}{dt} = (X_5 - X_2)X_3 - X_4 + 8,$ $\frac{dX_5}{dt} = (X_6 - X_3)X_4 - X_5 + 8, \frac{dX_6}{dt} = (X_1 - X_4)X_5 - X_6 + 8$ |
| BSL(Ours) | $\frac{dX_1}{dt} = 1.0033(\pm2.37 \times 10^{-4})X_2X_6 - 0.9926(\pm2.03 \times 10^{-4})X_5X_6,$ $-0.9991(\pm9.04 \times 10^{-4})X_1 + 7.9773(\pm2.08 \times 10^{-2}),$ $\frac{dX_2}{dt} = 0.9963(\pm5.54 \times 10^{-5})X_1X_3 - 0.9942(\pm2.47 \times 10^{-4})X_1X_6$ $-1.0054(\pm2.31 \times 10^{-4})X_2 + 7.8719(\pm7.05 \times 10^{-4}),$ $\frac{dX_3}{dt} = 1.0106(\pm2.19 \times 10^{-4})X_2X_4 - 1.0029(\pm2.03 \times 10^{-4})X_1X_2$ $-0.9979(\pm5.3 \times 10^{-4})X_3 + 7.9938(\pm1.48 \times 10^{-2}),$ $\frac{dX_4}{dt} = 1.0067(\pm1.01 \times 10^{-4})X_3X_5 - 1.0103(\pm9.09 \times 10^{-5})X_2X_3$ $-0.9926(\pm2.36 \times 10^{-4})X_4 + 8.055(\pm2.41 \times 10^{-3}),$ $\frac{dX_5}{dt} = 0.9953 \pm (4.39 \times 10^{-4})X_4X_6 - 0.9922 \pm (3.88 \times 10^{-4})X_3X_4$ $-1.0072 \pm (4.1 \times 10^{-4})X_5 + 7.8538(\pm2.1 \times 10^{-3}),$ $\frac{dX_6}{dt} = 1.0095(\pm2.71 \times 10^{-4})X_1X_5 - 0.9967(\pm2.83 \times 10^{-4})X_4X_5$ $-0.9772(\pm2.46 \times 10^{-4})X_6 + 7.9964(\pm4.15 \times 10^{-2})$ |
| PINN-SR | N/A |
| SINDy | $\frac{dX_1}{dt} = 8.0985 - 0.9423X_1 - 0.1007X_4 - 0.1659X_6 + 0.9582X_2X_6 - 0.944X_5X_6$ $\frac{dX_2}{dt} = 7.9372 - 0.1268X_1 - 0.9607X_2 + 0.9636X_1X_3 - 0.9558X_1X_6$ $\frac{dX_3}{dt} = 8.1523 - 0.0983X_1 - 0.1652X_2 - 0.9952X_3 - 0.9541X_1X_2 + 0.9651X_2X_4$ $\frac{dX_4}{dt} = 7.4958 - 0.9754X_4 + 0.1515X_5 - 0.973X_2X_3 + 0.9206X_3X_5,$ $\frac{dX_5}{dt} = 7.8556 - 0.1146X_4 - 0.9457X_5 - 0.9559X_3X_4 + 1.0023X_4X_6$ $\frac{dX_6}{dt} = 7.6026 + 0.0934X_2 - 0.927X_6 + 0.9711X_1X_5 - 0.9754X_4X_5$ |
| RVM | $\frac{dX_1}{dt} = 0.9613(\pm1.21)X_2X_6 - 0.962(\pm1.27)X_5X_6,$ $-0.8983(\pm2.67)X_1 + 7.518(\pm5.88),$ $\frac{dX_2}{dt} = 0.9578(\pm1.31)X_1X_3 - 0.9693(\pm1.24)X_1X_6$ $-0.9217(\pm2.8)X_2 + 7.6442(\pm6.1575),$ $\frac{dX_3}{dt} = 0.9535(\pm1.48)X_2X_4 - 0.9803(\pm1.38)X_1X_2$ $-0.9468(\pm2.96)X_3 + 7.5323(\pm6.56),$ $\frac{dX_4}{dt} = 0.9365(\pm1.24)X_3X_5 - 0.9748(\pm1.15)X_2X_3$ $-0.9114(\pm2.59)X_4 + 7.67(\pm5.49),$ $\frac{dX_5}{dt} = 0.9981 \pm (1.26)X_4X_6 - 0.9667 \pm (1.2)X_3X_4$ $-0.9216 \pm (2.75)X_5 + 7.6146(\pm5.96),$ $\frac{dX_6}{dt} = 0.9677(\pm1.42)X_1X_5 - 0.9757(\pm1.38)X_4X_5$ $-0.8986(\pm2.86)X_6 + 7.6657(\pm6.57)$ |

**Table 5:** Analytical forms of unsteady PDE

| Name | $\epsilon$ (large) |
|---|---|
| | Advection Equation |
| True | $u_t = -u_x$ |
| BSL(Ours) | $u_t = -0.9988(\pm0.024)u_x$ |
| PINN-SR | $u_t = -0.997u_x$ |
| SINDy | $u_t = -0.9961u_x$ |
| RVM | $u_t = -0.5148(\pm0.106)u_x - 0.9797(\pm0.384)uu_x$ |
| | $-0.018(\pm0.208)u^2u_x - 0.075(\pm0.148)u^2u_{xxx} + 0.049(\pm0.111)u^3u_{xxx}$ |
| | Burgers' Equation |
| True | $u_t = -uu_x + 0.5u_{xx}$ |
| BSL(Ours) | $u_t = -0.9929(\pm0.086)uu_x + 0.4993(\pm0.005)u_{xx}$ |
| PINN-SR | $u_t = -1.0103uu_x + 0.5051u_{xx}$ |
| SINDy | $u_t = -0.8179uu_x$ |
| RVM | $-0.0809(\pm0.041)u_x - 1.6684(\pm0.2618)uu_x + 4.1835(\pm0.571)u^2u_x$ |
| | $-3.9068(\pm0.391)u^3u_x + 0.1916(\pm0.064)uu_{xx}$ |
| | $-1.0314(\pm0.197)u^2u_{xx} + 1.5504(0.156)u^3u_{xx}$ |
| | Burgers' Equation with Source |
| True | $u_t = -uu_x + 0.1u_{xx} + sin(x)sin(t)$ |
| BSL(Ours) | $-0.9882(\pm0.246)uu_x + 0.105(\pm0.022)u_{xx} + 0.9859(\pm0.005)sin(x)sin(t)$ |
| PINN-SR | $u_t = -0.9576uu_x + 0.1168u_{xx} + 1.0179sin(x)sin(t)$ |
| SINDy | $u_t = 0.8052sin(x)sin(t)$ |
| RVM | $-0.0234(\pm0.145)uu_x + 0.8318(\pm0.142)sin(x)sin(t)$ |
| | $-0.0789(\pm0.105)sin(x) + 0.3558(\pm0.156)sin(x)cos(t)$ |

**Table 6:** Analytical forms of steady PDE

| Name | $\epsilon$ (large) |
|---|---|
| | Heat Equation |
| True | $u_{yy} = -u_{xx}$ |
| BSL(Ours) | $u_{yy} = -0.9611(\pm0.059)u_{xx}$ |
| PINN-SR | $u_{yy} = 0.5544u_x$ |
| SINDy | $u_{yy} = -0.069u_{xx} + 13.8988uu_x - 19.493u_x + 0.2468uu_{xx}$ |
| RVM | $u_{yy} = -7.5861(\pm202.6)u_x$ |
| | Poisson Equation |
| True | $u_{yy} = -u_{xx} - sin(x)sin(y)$ |
| BSL(Ours) | $u_{yy} = -0.9788(\pm8.75\times10^{-4})u_{xx} - 0.9897(\pm4.63\times10^{-4})sin(x)sin(y)$ |
| PINN-SR | $u_{yy} = 0.13611752uu_x + 0.29748484uu_{xxx}$ |
| SINDy | $u_{yy} = 0.2316u_{xx} - 0.4221sin(x)sin(y)$ |
| RVM | $u_{yy} = 0.2284(\pm0.007)u_{xx} - 0.3954(\pm0.01)sin(x)sin(y)$ |

## A.9 Implementation detail for the spline

In the current work, we apply direct tensor product to extend spline for solving spatial-temporal field and the relevant statistics are attached in Tab. 7, where the numerber of control points (trainable weights) $\theta$ in 1-d is marked by red and the total number of control points for the 2-d scenario is listed in the last column. We only store the non-zero elements for two-dimensional basis to leverage the sparsity (local support) of the spline. However, we must claim that it is *not* an optimal way to extend spline for higher spatial dimensions. In that case, a spline kernel can be defined and the tensor-product is only processed in the subdomain, as shown in [43].

**Table 7:** Direct tensor-product spline for PDE

| basis $x$ | basis $t(y)$ | basis$(x, t(y))$ | sparsity | trainable params |
|---|---|---|---|---|
| | | Advection Equation | | |
| $50 \times 54$ | $50 \times 54$ | 62500 | 0.0086 | 2916 |
| | | Burgers' Equation | | |
| $128 \times 13$ | $101 \times 19$ | 92872 | 0.058 | 247 |
| | | Burgers' Equation with Source | | |
| $201 \times 103$ | $101 \times 103$ | 251415 | 0.0012 | 10609 |
| | | Heat Equation | | |
| $51 \times 11$ | $51 \times 11$ | 41209 | 0.1309 | 121 |
| | | Poisson | | |
| $101 \times 53$ | $101 \times 53$ | 162409 | 0.0057 | 2809 |

## A.10 Relevant Terminologies

**Table 8:** Terminologies

| Name | Symbol | Explanation |
|---|---|---|
| Control points | $\theta$ | Trainable weight $\theta$ for spline basis. |
| Knots | $\tau_s$ | Location of control points. |
| Measurement points | | Sparse spatio-temporal points with labels. |
| Collocation points | | Dense spatio-temporal points without labels. |
| | $\mathbf{N_m}$ | Spline basis evaluated at measurement points. |
| | $\mathbf{N_c}$ | Spline basis evaluated at collocation points. |
| Library candidates | $\mathbf{\Phi}$ | A collection of polynomial terms that the system identification algorithm can choose parsimonious terms from it. e.g., $\{x, y, x^2y, ...\}$ (for ODE) or $\{u, uu_x, u_{xx}...\}$ for (PDE). |
| ADO iteration | | Alternating direction optimization to update the trainable parameters including control points $\theta$, weight of library candidates $\mathbf{W}$ and covariance matrices. |
| Aleatoric Uncertainty | | Due to intrinsic randomness by nature, which is irreducible. |
| Epistemic Uncertainty | | Because of a lack of knowledge, which can be reduced by adding more information. |

## A.11 Real world application: predator-prey system

In this section, we would test our proposed BSL model on one real-world case, predator-prey system. The real data set is obtained online and it depicts the population of hares and lynx from 1900 to 1920 from Hudson Bay Company. The data is presented in Tab. 11: The reference governing equation by mathematical analysis is:

$$\frac{dx}{dt} = 0.4807x - 0.0248xy \tag{28}$$

$$\frac{dy}{dt} = -0.9272y + 0.0276xy \tag{29}$$

We test the 4 methods on this data set and the result can be found in Tab 9 .We have made assumptions about constructing the libraries. We assume the predator (lynx) only feeds on the prey (hares). Meantime, the prey (hares) only has one predator (lynx). Therefore, the change rate of these two species can only depend on themselves ($x$,$y$) and some higher order correlations between them ($xy, x^2y, xy^2$). The discovered forms of the 4 methods are listed in Tab. 10. Finally, the UQ results are shown in Fig 11. In short, only the proposed method work on the real sparse and noisy dataset. And the UQ prediction covers more measurement points than the reference model Eq. 28, which helps to better explain the data set.

**Table 9:** ODE discovery comparison

| Name | **rmse** (large) | $\mathbf{M_P}$ | $\mathbf{M_R}$ | Training Cost |
|---|---|---|---|---|
| | | Predator-prey | | |
| **BSL(Ours)** | **30**.4 | **1** | **1** | $\sim 2278(+6)s$ |
| PINN-SR | Fail | 0.5 | 0.25 | $\sim 2988s$ |
| SINDy | Fail | 0.6 | 0.75 | $\sim 10s$ |
| RVM | Fail | 0.8 | 1 | $\sim 10s$ |

**Table 10:** Analytical forms of ODE

| Name | $\epsilon$ (large) |
|---|---|
| | Predator-prey (Lotka-Volterra) |
| True | $\frac{dx}{dt} = 0.4807x - 0.0248xy, \frac{dy}{dt} = -0.9272y + 0.0276xy$ |
| BSL(Ours) | $\frac{dx}{dt} = -0.5124(\pm 0.028)x - 0.0266(\pm 8.72 \times 10^{-4})xy$ $\frac{dy}{dt} = -0.9258(\pm 0.065)y + 0.0279(\pm 1.47 \times 10^{-3})xy$ |
| PINN-SR | $\frac{dx}{dt} = -13.9238y$ $\frac{dy}{dt} = -0.1144y$ |
| SINDy | $\frac{dx}{dt} = 0.5813x - 0.0261xy,$ $\frac{dy}{dt} = 0.2549x - 0.2702y$ |
| RVM | $\frac{dx}{dt} = 0.5732(\pm 0.6488)x - 0.2386(\pm 0.4643)y - 0.0253(\pm 0.1432)xy,$ $\frac{dy}{dt} = -0.8018(\pm 0.9459)y + 0.0226(\pm 0.1481)xy$ |

**Table 11:** Lynx-Hares population

| Year | Hares($\times 1000$) | Lynx($\times 1000$) |
|---|---|---|
| 1900 | 30 | 4 |
| 1901 | 47.2 | 6.1 |
| 1902 | 70.2 | 9.8 |
| 1903 | 77.4 | 35.2 |
| 1904 | 36.3 | 59.4 |
| 1905 | 20.6 | 41.7 |
| 1906 | 18.1 | 19 |
| 1907 | 21.4 | 13 |
| 1908 | 22 | 8.3 |
| 1909 | 25.4 | 9.1 |
| 1910 | 27.1 | 7.4 |
| 1911 | 40.3 | 8 |
| 1912 | 57 | 12.3 |
| 1913 | 76.6 | 19.5 |
| 1914 | 52.3 | 45.7 |
| 1915 | 19.5 | 51.1 |
| 1916 | 11.2 | 29.7 |
| 1917 | 7.6 | 15.8 |
| 1918 | 14.6 | 9.7 |
| 1919 | 16.2 | 10.1 |
| 1920 | 24.7 | 8.6 |

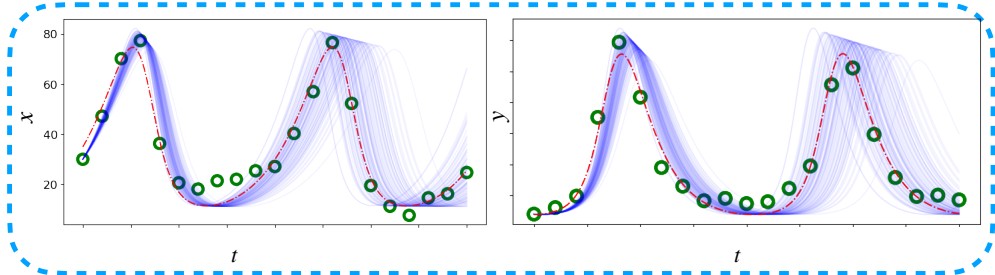

**Figure 11:** UQ results for predator-prey system; left sub-fig shows the result for state $x$, the hares population; the right sub-fig shows the result for state $y$, the lynx population. The green dot is the sparse and biased measurement data. The red line is the reference model and the blue curves are the ensemble predictions from our proposed model.

### A.12 Experiment on different smoothing algorithms

This section we will test the effect of different smoothing algorithms and its impact to the SINDy result. These methods include smoothing based polynomial interpolation, convolutional smoother, smoothing with Tikhonov regularization and smoothing with spline fitting. We performed comparison of SINDy with these smoothing methods on a representative ODE system (Van der Pol system) and PDE system (Poisson equation) studied in this work. As shown in the Table 12 and Table 13 below, when the data noise is above $5\%$, although data is preprocessed using smoothing and uniform resampling, none of these method work. Basically, the SINDy still failed to discover the correct model forms with different smoothing schemes, and the identified systems are different from the true. In contrast, our proposed approach is very robust and superior to handling corrupted data, thanks to the spline learning in Bayesian settings.

**Table 12:** ODE and PDE discovery comparison

| Name | **rmse**($\epsilon = 5\%$) | $\mathbf{M_P}$ | $\mathbf{M_R}$ | Training Cost |
|---|---|---|---|---|
| | Van der Pol Oscillator | | | |
| **BSL(Ours)** | **18.04** | **1** | **1** | $\sim 133(+3)s$ |
| SINDy(No smoother) | Fail | 0.33 | 1 | $\sim 10s$ |
| SINDy(Poly) | Fail | 0.6 | 0.75 | $\sim 10s$ |
| SINDy(Conv) | Fail | 0.4 | 1 | $\sim 10s$ |
| SINDy(Tikhonov) | Fail | 0.21 | 1 | $\sim 10s$ |
| SINDy(Spline) | Fail | 0.25 | 1 | $\sim 10s$ |
| | Poisson Equation | | | |
| **BSL(Ours)** | **16.7** | **1** | **1** | $\sim 92(+9)s$ |
| SINDy(No smoother) | Fail | 0.5 | 1 | $\sim 10s$ |
| SINDy(Poly) | 962 | **1** | 1 | $\sim 10s$ |
| SINDy(Conv) | Fail | 0.66 | 1 | $\sim 10s$ |
| SINDy(Tikhonov) | Fail | 0.5 | 1 | $\sim 10s$ |
| SINDy(Spline) | Fail | 0.5 | 1 | $\sim 10s$ |

**Table 13:** Smoothing algorithm effects

| Name | Analytical form |
|---|---|
| | Van der Pol Oscillator |
| True | $\frac{dx}{dt} = y$ 
 $\frac{dy}{dt} = -x - 0.5x^2y + 0.5y$ |
| SINDy(No smoother) | $\frac{dx}{dt} = 0.0277 - 0.217x + 1.4y - 0.025x^2$ 
 $+0.0542x^3 - 0.1285x^2y + 0.1039xy^2 - 0.0896y^3,$ 
 $\frac{dy}{dt} = -0.9835x + 0.3462y - 0.4675x^2y + 0.0325y^3$ |
| SINDy(Poly) | $\frac{dx}{dt} = 0.1197 + 0.169x + 0.9975y - 0.0292x^2$ 
 $+0.0232xy - 0.0274y^2 - 0.0393x^3 + 0.0208x^2y - 0.0463xy^2,$ 
 $\frac{dy}{dt} = -1.0909x + 0.149y + 0.0294x^3 - 0.4x^2y - 0.0356xy^2 + 0.0762y^3$ |
| SINDy(Conv) | $\frac{dx}{dt} = 0.1518 + 0.9978y - 0.0486x^2 + 0.0274xy - 0.0327y^2,$ 
 $\frac{dy}{dt} = -1.1154x - 0.2143y + 0.0348x^3 - 0.4374x^2y + 0.0613y^3$ |
| SINDy(Tikhonov) | $\frac{dx}{dt} = -0.2572 + 0.1486x + 1.1299y + 0.0982x^2 - 0.0721xy,$ 
 $+0.0549y^2 - 0.0478x^2y - 0.0816xy^2 - 0.0542y^3$ 
 $\frac{dy}{dt} = 0.2338 - 1.3282x + 0.1603y - 0.0718x^2$ 
 $+0.058xy - 0.0564y^2 + 0.1069x^3 - 0.0842x^2y + 0.1373xy^2 - 0.0415y^3$ |
| SINDy(Spline) | $\frac{dx}{dt} = 0.1916 + 1.5304y - 0.0614x^2 + 0.0215xy$ 
 $-0.0325y^2 - 0.155x^2y + 0.0796xy^2 - 0.1168y^3$ 
 $\frac{dy}{dt} = -0.35 - 1.0127x + 0.6756y + 0.0695x^2$ 
 $+0.0741y^2 - 0.5493x^2y + 0.0523xy^2 - 0.0412y^3$ |
| | Poisson equation |
| True | $u_{yy} = -u_{xx} - sin(x)sin(y)$ |
| SINDy(No smoother) | $u_{yy} = 0.5635u_{xx} + 1.683uu_x + 0.088uu_{xx} - 0.17sin(x)sin(y)$ |
| SINDy(Poly) | $u_{yy} = 0.2316u_{xx} - 0.4221sin(x)sin(y)$ |
| SINDy(Conv) | $u_{yy} = 0.47u_{xx} - 0.2649sin(x)sin(y) + 0.073sin(x)cos(y)$ |
| SINDy(Tikhonov) | $u_{yy} = -0.17u_{xx} - 0.087uu_x - 0.458sin(x)sin(y) + 0.02sin(x)cos(y)$ |
| SINDy(Spline) | $u_{yy} = 0.1562u_{xx} + 0.1751uu_x - 0.3951sin(x)sin(y) - 0.0911sin(x)cos(y)$ |