# OpenReview forum: "Bayesian Spline Learning for Equation Discovery of Nonlinear Dynamics with Quantified Uncertainty"
_NeurIPS.cc/2022/Conference — NeurIPS 2022 Accept_

### Official Review · Reviewer_3NV7 · 2022-07-02

**Rating:** 5
**Confidence:** 3
**Soundness:** 2 fair
**Presentation:** 3 good
**Contribution:** 2 fair

**Summary:**

The authors propose a Bayesian Spline Learning method for equation discovery, which uses spline basis to handle the data scarcity and noise. The Bayesian uncertainty calibration techniques are used to approximate posterior distributions of the trainable parameters. The proposed method is built upon the ADO of PINN-SR, but outperforms PINN-SR and SINDy in all the metrics in the experiments of four equations.

**Questions:**

In the current presentation, it can still be hard to interpret why uncertainty calibration can benefit equation discovery against measurement noise and sparsity. We only know that the spline learning can perform differentiation accurately to enhance learning efficiency and the posterior distribution of spline, library and equation coefficients are estimated simultaneously. These explain the training time superiority but lacks in interpretation why the obtained uncertainty can benefit equation discovery/coefficient estimation, which is more at the core of the question. It makes it harder to understand why the proposed method has a smaller error than PINN-SR and SINDy, especially calibration method generally succeeds in efficiency of uncertainty estimation but hardly provides significant accuracy boost based on relevant studies of previous works on other topics such as traffic prediction, etc.

**Limitations:**

The authors mention the limitation of pre-defined library of candidate terms.

**Strengths And Weaknesses:**

Strengths:
- The motivation is clear and the used method makes sense in the current presentation.
- The experimental results support the claim that their method is more robust with 10% data with large noise (5% to 20%) on Van der Pol Oscillator, Lorenz 96, Advection Equation and Burgers' Equation.
- The codes and theoretical properties of the algorithm are provided in the supplementary materials.

Weaknesses:
- It is not clear what are the analytical forms of the used four equations and what are the parsimonious equations discovered. There should be some descriptions of the discovered equations at least in the supplementary materials if not in the main manuscript.
- The presentation about why the propose method can benefit equation discovery can be further extended and described in detail.

---

> ### Author Response · Authors · 2022-08-02
> **Response to Reviewer 4: Part 1**
>
> ### Summary
>
> > 1. The authors propose a Bayesian Spline Learning method for equation discovery, which uses spline basis to handle the data scarcity and noise. The Bayesian uncertainty calibration techniques are used to approximate posterior distributions of the trainable parameters. The proposed method is built upon the ADO of PINN-SR, but outperforms PINN-SR and SINDy in all the metrics in the experiments of four equations.
>
> ***Response:***
> We sincerely thank the reviewer's overall assessment and appreciate his/her constructive comments and suggestions, which are very helpful for improving our paper. Please find our responses below. Revisions have also been made in the paper.
>
> ### Strengths And Weaknesses:
>
> > 1. It is not clear what are the analytical forms of the used four equations and what are the parsimonious equations discovered. There should be some descriptions of the discovered equations at least in the supplementary materials if not in the main manuscript.
>
> ***Response:***
> We apologize for the lack of details of reference and discovered analytical forms, which has been addressed in the revised manuscript. Following reviewer's suggestion, we added a new section in the supplementary materials to provide the analytical forms of the four ODEs/PDEs and the results of discovered parsimonious equations. Please see Appendix A.8 in supplementary materials.
>
>
> > 2. The presentation about why the proposed method can benefit equation discovery can be further extended and described in detail.
>
> ***Response:***
> We thank the reviewer for this comment and suggestion. The benefit of the proposed method for equation discovery is comprehensively discussed in the Response to Questions below. Moreover, we extended the discussion section in the revised manuscript to further describe it in detail.

---

> > ### Author Response · Authors · 2022-08-02
> > **Response to Reviewer 4: Part 2**
> >
> > ### Questions
> >
> > > 1. In the current presentation, it can still be hard to interpret why uncertainty calibration can benefit equation discovery against measurement noise and sparsity. We only know that spline learning can perform differentiation accurately to enhance learning efficiency and the posterior distribution of spline, library and equation coefficients are estimated simultaneously. These explain the training time superiority but lacks in interpretation why the obtained uncertainty can benefit equation discovery/coefficient estimation, which is more at the core of the question. It makes it harder to understand why the proposed method has a smaller error than PINN-SR and SINDy, especially the calibration method generally succeeds in the efficiency of uncertainty estimation but hardly provides significant accuracy boost based on relevant studies of previous works on other topics such as traffic prediction, etc.
> >
> > ***Response:***
> > We thank the reviewer for raising this question. The Bayesian formulation of equation discovery under uncertainties has two benefits. First, it is more robust in scenarios with large data noise and sparsity, since Bayesian learning is able to provide the whole posterior distributions of coefficients and the spread of all possible solutions would mitigate overfitting issues and is more naturally combined with the following Bayesian sequential threshold pruning step to find the most parsimonious form. This is in contrast to deterministic equation discovery schemes such as SINDy or PINN-SR, where the value of each coefficient is estimated deterministically, which is more likely to suffer from aliasing errors and overfitting issues when data is very noisy and sparse, leading to many spurious and redundant terms. Second, uncertainty quantification (UQ) itself is critical in most scientific and engineering applications. As for the equation discovery, substantial uncertainties can be introduced from observation noise, data sparsity, and inaccuracy of pre-defined libraries. Therefore, we believe quantifying the uncertainty is as important as discovering the equations. We agree with the reviewer that quantifying uncertainty itself cannot directly improve the accuracy of equation discovery/coefficient estimation, especially when data is sufficient and clean. Please note that one of the main reasons that the proposed BSL method has a smaller error than PINN-SR and SINDy is due to spline learning, which is capable of accurately recovering derivative information from noisy data.
> >
> > Although SINDy, PINN-SR or their variants can also be formulated in the Bayesian framework to quantify uncertainty, they cannot address all the challenges mentioned above simultaneously. The SINDy family is known to be less capable of handling large amounts of measurement noise, whereas PINN-SR fails to deal with ODE discovery and tends to have convergence issues under large data uncertainty. This work introduces spline learning to resolve the data noise issue, and the unified loss function is naturally extended to a likelihood function for Bayesian inference. It is not clear how to achieve these straightforwardly for SINDy or PINN-SR methods. To summarize, the UQ cannot directly improve the accuracy of coefficient estimation. However, it will improve the robustness of the algorithm and, more importantly, it can provide prediction confidence of the discovered models. In contrast to most deterministic methods, our probabilistic method will provide uncertainty distributions to reflect the certainty of the model discovery without introducing too much computation overhead.

---

> > > ### Author Response · Authors · 2022-08-02
> > > **Response to Reviewer 4: Part 3 Addtional tables for the identified systems (ODE)**
> > >
> > > |Analytical ODE | |
> > > | :---: | :---: |
> > > | Name | $\epsilon$ (large) |
> > > |  | Vander Pol Oscillator |
> > > | True | $\frac{d x}{d t}=y, \frac{d y}{d t}=-x-0.5 x^{2} y+0.5 y$ |
> > > | BSL(Ours) | $\frac{d x}{d t}=1.0096(\pm 0.024) y, \frac{d y}{d t}=-0.9858(\pm 0.037) x-0.4801(\pm 0.114) x^{2} y+0.4889(\pm 0.111) y$ |
> > > | PINN-SR | $\frac{dx}{dt} = 1.3079-0.1151x+0.9982y+0.1939x^2-0.4101xy+0.8559y^2-0.3577x^3$|
> > > | |$+0.6764x^2y-0.1207xy^2+0.6261y^3$|
> > > | |$\frac{dy}{dt}=-0.6718x+1.6035y+1.6339y^2+0.4803y^3$|
> > > |SINDy |$\frac{dx}{dt}=0.1197+0.169x+0.9975y-0.0292x^2+0.0232xy-0.0274y^2-0.0393x^3+0.0208x^2y-0.0463xy^2$|
> > > ||$\frac{dy}{dt} = -1.0909x+0.149y+0.0294x^3-0.4x^2y-0.0356xy^2+0.0762y^3$|
> > > |RVM|$\frac{dx}{dt}=0.9957(\pm0.073)y$,$\frac{dy}{dt} = -0.9943(\pm0.0597)x-0.4777(\pm0.0809)x^2y+0.4714(\pm0.1204)y$|
> > > |  | Lorenz 96 |
> > > |True|$\frac{dX_1}{dt} = (X_2-X_5)X_6-X_1+8$, $\frac{dX_2}{dt} = (X_3-X_6)X_1-X_2+8$|
> > > |    |$\frac{dX_3}{dt} = (X_4-X_1)X_2-X_3+8$, $\frac{dX_4}{dt} = (X_5-X_2)X_3-X_4+8$,|
> > > |    |$\frac{dX_5}{dt} = (X_6-X_3)X_4-X_5+8$, $\frac{dX_6}{dt} = (X_1-X_4)X_5-X_6+8$|
> > > |BSL(Ours)| $\frac{dX_1}{dt} = 1.0033(\pm2.37\times10^{-4})X_2X_6-0.9926(\pm2.03\times10^{-4})X_5X_6$|
> > > |    |$-0.9991(\pm9.04\times10^{-4})X_1+7.9773(\pm2.08\times10^{-2})$|
> > > |    |$\frac{dX_2}{dt} = 0.9963(\pm5.54\times10^{-5})X_1X_3-0.9942(\pm2.47\times10^{-4})X_1X_6$|
> > > |    |$-1.0054(\pm2.31\times10^{-4})X_2+7.8719(\pm7.05\times10^{-4})$|
> > > |    |$\frac{dX_3}{dt} = 1.0106(\pm{2.19\times10^{-4}})X_2X_4-1.0029(\pm{2.03\times10^{-4}})X_1X_2$|
> > > |    |$-0.9979(\pm5.3\times10^{-4})X_3+7.9938(\pm{1.48\times10^{-2}})$|
> > > |    |$\frac{dX_4}{dt} = 1.0067(\pm1.01\times10^{-4})X_3X_5-1.0103(\pm9.09\times10^{-5})X_2X_3$|
> > > |    |$-0.9926(\pm2.36\times10^{-4})X_4+8.055(\pm{2.41\times10^{-3}})$|
> > > |    |$\frac{dX_5}{dt} = 0.9953\pm(4.39\times10^{-4})X_4X_6-0.9922\pm(3.88\times10^{-4})X_3X_4$|
> > > |    |$-1.0072\pm(4.1\times10^{-4})X_5+7.8538(\pm{2.1\times10^{-3}})$|
> > > |    |$\frac{dX_6}{dt} = 1.0095(\pm2.71\times10^{-4})X_1X_5-0.9967(\pm2.83\times10^{-4})X_4X_5$|
> > > |    |$-0.9772(\pm2.46\times10^{-4})X_6+7.9964(\pm4.15\times10^{-2})$|
> > > |PINN-SR| N/A|
> > > |SINDy |$\frac{dX_1}{dt} = 8.0985-0.9423X_1-0.1007X_4-0.1659X_6+0.9582X_2X_6-0.944X_5X_6$|
> > > |    |$\frac{dX_2}{dt} = 7.9372-0.1268X_1-0.9607X_2+0.9636X_1X_3-0.9558X_1X_6$|
> > > |    |$\frac{dX_3}{dt} = 8.1523-0.0983X_1-0.1652X_2-0.9952X_3-0.9541X_1X_2+0.9651X_2X_4$|
> > > |    |$\frac{dX_4}{dt} = 7.4958-0.9754X_4+0.1515X_5-0.973X_2X_3+0.9206X_3X_5$|
> > > |    |$\frac{dX_5}{dt} = 7.8556-0.1146X_4-0.9457X_5-0.9559X_3X_4+1.0023X_4X_6$|
> > > |    |$\frac{dX_6}{dt} = 7.6026+0.0934X_2-0.927X_6+0.9711X_1X_5-0.9754X_4X_5$|
> > > |RVM| $\frac{dX_1}{dt} = 0.9613(\pm1.21)X_2X_6-0.962(\pm1.27)X_5X_6$|
> > > |    |$-0.8983(\pm2.67)X_1+7.518(\pm5.88)$|
> > > |    |$\frac{dX_2}{dt} = 0.9578(\pm1.31)X_1X_3-0.9693(\pm1.24)X_1X_6$|
> > > |    |$-0.9217(\pm2.8)X_2+7.6442(\pm6.1575)$
> > > |    |$\frac{dX_3}{dt} = 0.9535(\pm{1.48})X_2X_4-0.9803(\pm{1.38})X_1X_2$|
> > > |    |$-0.9468(\pm2.96)X_3+7.5323(\pm6.56)$|
> > > |    |$\frac{dX_4}{dt} = 0.9365(\pm1.24)X_3X_5-0.9748(\pm1.15)X_2X_3$|
> > > |    |$-0.9114(\pm2.59)X_4+7.67(\pm5.49)$|
> > > |    |$\frac{dX_5}{dt} = 0.9981\pm(1.26)X_4X_6-0.9667\pm(1.2)X_3X_4$|
> > > |    |$-0.9216\pm(2.75)X_5+7.6146(\pm5.96)$|
> > > |    |$\frac{dX_6}{dt} = 0.9677(\pm1.42)X_1X_5-0.9757(\pm1.38)X_4X_5$|
> > > |    |$-0.8986(\pm2.86)X_6+7.6657(\pm6.57)$|
> > > || |

---

> > > > ### Author Response · Authors · 2022-08-02
> > > > **Response to Reviewer 4: Part 4 Addtional tables for the identified systems (PDE)**
> > > >
> > > > |Analytical forms of PDE | |
> > > > | :---: | :---: |
> > > > | Name | $\epsilon$ (large) |
> > > > |  | Advection Equation |
> > > > |True|$u_t=-u_x$|
> > > > |BSL(Ours)|   $u_t = -0.9988(\pm0.024)u_x$|
> > > > |PINN-SR|$u_t=-0.997u_x$|
> > > > |SINDy |$u_t=-0.9961u_x$|
> > > > |RVM|$u_t = -0.5148(\pm0.106)u_x-0.9797(\pm0.384)uu_x$|
> > > > |  |$-0.018(\pm0.208)u^2u_x-0.075(\pm0.148)u^2u_{xxx}+0.049(\pm0.111)u^3u_{xxx}$|
> > > > |  | Burgers' Equation |
> > > > |True|$u_t = -uu_x+0.5u_{xx}$|
> > > > |BSL(Ours)|  $u_t = -0.9929(\pm0.086)uu_x+0.4993(\pm0.005)u_{xx}$|
> > > > |PINN-SR|$u_{t} = -1.0103uu_{x}+0.5051u_{xx}$|
> > > > |SINDy |$u_t=-0.8179uu_x$|
> > > > |RVM|$-0.0809(\pm0.041)u_x-1.6684(\pm0.2618)uu_x+4.1835(\pm0.571)u^2u_x$|
> > > > |  |$-3.9068(\pm0.391)u^3u_x+0.1916(\pm0.064)uu_{xx}-1.0314(\pm0.197)u^2u_{xx}+1.5504(0.156)u^3u_{xx}$|
> > > > |  | Burgers' Equation with Source |
> > > > |True|$u_t = -uu_x+0.1u_{xx}+sin(x)sin(t)$|
> > > > |BSL(Ours)|  $-0.9882(\pm0.246)uu_x+0.105(\pm0.022)u_{xx}+0.9859(\pm0.005)sin(x)sin(t)$|
> > > > |PINN-SR|$u_t = -0.9576uu_x+0.1168u_{xx}+1.0179sin(x)sin(t)$|
> > > > |SINDy |$u_t =0.8052sin(x)sin(t)$|
> > > > |RVM|$-0.0234(\pm0.145)uu_x+0.8318(\pm0.142)sin(x)sin(t)$|
> > > > |  |$-0.0789(\pm0.105)sin(x)+0.3558(\pm0.156)sin(x)cos(t)$|
> > > > |  | Heat Equation |
> > > > |True|$u_{yy}=-u_{xx}$|
> > > > |BSL(Ours)|  $u_{yy}=-0.9611(\pm0.059)u_{xx}$|
> > > > |PINN-SR|$u_{yy}=0.5544u_x$|
> > > > |SINDy |$u_{yy}=-0.069u_{xx}+13.8988uu_x-19.493u_x+0.2468uu_{xx}$|
> > > > |RVM|$u_{yy}=-7.5861(\pm202.6)u_{x}$|
> > > > |  | Poisson Equation|
> > > > |True|$u_{yy} = -u_{xx}-sin(x)sin(y)$|
> > > > |BSL(Ours)|  $u_{yy} = -0.9788(\pm8.75\times10^{-4})u_{xx}-0.9897(\pm4.63\times10^{-4})sin(x)sin(y)$|
> > > > |PINN-SR|$u_{yy} = 0.13611752uu_x+0.29748484uu_{xxx}$|
> > > > |SINDy |$u_{yy}=0.2316u_{xx}-0.4221sin(x)sin(y)$|
> > > > |RVM|$u_{yy}=0.2284(\pm0.007)u_{xx}-0.3954(\pm0.01)sin(x)sin(y)$|
> > > > || |

---

> ### Author Response · Authors · 2022-08-08
> **Thank you for your insightful comments**
>
> Dear reviewer 3NV7,
>
> Again, we would like to thank your insightful comments on our paper. We believe we have fully addressed your concerns through our response earlier and the revised paper. We do hope to have your feedback and look forward to answering any additional questions you have. Thank you very much.

---

### Official Review · Reviewer_nVYU · 2022-07-07

**Rating:** 6
**Confidence:** 4
**Soundness:** 3 good
**Presentation:** 3 good
**Contribution:** 2 fair

**Summary:**

This paper extended an existing spline-based model for fitting dynamic systems (ODEs) reported in [14]. There are two main changes: 1) Use tensor-product of splines to model both time and spatial dimensions. 2) Use Bayesian regression to fit the coefficients. With the first change, authors attempted to model PDEs with one spatial dimension (avection equation and Burgers' equation).

The authors also claimed that the Bayesian estimates help with the data assimilation tasks, although the approach was not described in detail.



**Questions:**

1. Equation (3), for PDE, how do you model terms like $u_x$ and $u_{xx}$? Please give an example like Burger's equation. Is the representation in $R^{n\times s\times m}$?
2. Equation (10), what is $U$? Same for equation (13).
3. Table 1, what are the values?


**Limitations:**

The authors should clarify the limitations of splines in modeling higher-dimension data.

**Strengths And Weaknesses:**

Strength:

The paper contains some reasonable ideas (tricks) that have not seemed to be explored before.

Weaknessses:

1. The paper contains quite a few typos in equations and counter-intuitive use of symbols (e.g., using $\hat{U}$ to represent data instead of an estimate). These problems can make the method difficult to understand. The author should have been more careful proof-reading the draft before submission.
2. The tensor-product of 1-d splines has severe limitations, particularly when the spatial dimension is higher. The authors should have made this limitation more clear.
3. The experiment results do not contain sufficient comparison with baseline methods (Only Table 1 contains the comparison of, I assume, the running time).

---

> ### Author Response · Authors · 2022-08-02
> **Response to Reviewer 3: Part 1**
>
> ### Summary:
>
> > 1. This paper extended an existing spline-based model for fitting dynamic systems (ODEs) reported in [14]. There are two main changes: 1) Use tensor-product of splines to model both time and spatial dimensions. 2) Use Bayesian regression to fit the coefficients. With the first change, authors attempted to model PDEs with one spatial dimension (avection equation and Burgers' equation).
> The authors also claimed that the Bayesian estimates help with the data assimilation tasks, although the approach was not described in detail.
>
> ***Response:*** We sincerely thank the reviewer's overall assessment and appreciate his/her constructive comments and suggestions, which are very helpful for improving our paper. Please find our responses below. Revisions have also been made in the paper.
>
> In the existing literature, most state-of-the-art data-driven interpretable learning methods mainly use either finite difference (FD) or auto differentiation (AD) to compute derivatives, which is critical for explicit model form discovery. However, the FD is very sensitive to data noise, while the AD approach generally does not work for ODE systems. The spline-based models have shown good potential to deal with corrupted datasets, as demonstrated in recent developments [1][2]. However, there are still significant gaps in handling data scarcity and noise, quantifying and reducing multi-source uncertainties, and promoting sparsity in both ODE/PDE discovery, which most existing techniques struggle to simultaneously address. This work bridges the gap and can simultaneously identify both ODE and PDE systems from corrupted datasets, meanwhile the associated uncertainty is well quantified without introducing much computational overhead. Moreover, theoretical analysis and error bounds of the proposed method are provided with rigorous proofs. We believe this work is an important contribution to the literature in the field of data-driven equation discovery.
>
> As the reviewer also noticed, the Bayesian formulation of spline learning is a critical component in this work, which improves the algorithm robustness, enables uncertainty quantification, and helps with the downstream data assimilation tasks since the model-form uncertainty information (e.g., state covariance), which is often unknown, can be determined rigorously and naturally used in Bayesian analysis step (e.g., Kalman update). More details about the data assimilation method, EnKF, are provided in Appendix A.3 of the supplementary materials.
>
> ***Reference***
>
> [1]. F. Sun, Y. Liu, H. Sun, Physics-informed spline learning for nonlinear dynamics discovery, in Proceedings of the Thirtieth International Joint Conference on Artificial Intelligence (IJCAI-21),2021, pp. 2054–2061.
>
> [2]. M. Fey, J. E. Lenssen, F. Weichert, H. Müller, Splinecnn: Fast geometric deep learning with
> continuous b-spline kernels, in Proceedings of the IEEE Conference on Computer Vision and
> Pattern Recognition, 2018, pp. 869–877.
>
>
>
> ### Strengths And Weaknesses:
>
> >1. The paper contains quite a few typos in equations and counter-intuitive use of symbols (e.g., using
>     $\hat{U}$ to represent data instead of an estimate). These problems can make the method difficult to understand. The author should have been more careful proof-reading the draft before submission.
>
> ***Response:*** We thank the reviewer for pointing out this issue. We have checked the paper thoroughly and revised the paper, especially on the typos and notations. For example, now we use $\hat{\cdot}$ to denote estimation and $\tilde{\cdot}$ to denote noisy observation to avoid any confusion. Please see the revised paper with track changes (marked in blue color).

---

> > ### Author Response · Authors · 2022-08-02
> > **Response to Reviewer 3: Part 2**
> >
> > > 2. The tensor-product of 1-d splines has severe limitations, particularly when the spatial dimension is higher. The authors should have made this limitation more clear
> >
> > ***response:*** We agree with the reviewer that the tensor-product of 1-D splines has the scalability limitation to high-dimensional problems. For most spatiotemporal PDEs, the highest spatial dimension is three, which is still manageable but with significantly increased computational cost. However, for high-dimensional PDEs (e.g., Hamilton-Jacobi-Bellman equation, Allen-Cahn equation, etc.), extension of splines using tensor-product is infeasible. We explicitly discussed this limitation in the revised manuscript. Moreover, as suggested by the reviewer R79g, there are some recent advances in spline learning to reduce the computational cost and improve the expressibility for high-dimensional cases. For instance, combining deep learning with continuous spline kernels [1] or using neural networks to optimize knot size [2] show great promise for lowering computational costs in high-dimensional settings. The extension of spline learning for high-dimensional problems will be explored in our future work.
> >
> > ***Reference***
> >
> > [1].M. Fey, J. E. Lenssen, F. Weichert, H. Müller, Splinecnn: Fast geometric deep learning with continuous b-spline kernels, in: Proceedings of the IEEE Conference on Computer Vision and
> > Pattern Recognition, 2018, pp. 869–877.
> >
> > [2].C. Durkan, A. Bekasov, I. Murray, G. Papamakarios, Neural spline flows, Advances in neural
> > information processing systems 32 (2019)
> >
> >
> >
> > > 3. The experiment results do not contain sufficient comparison with baseline methods (Only Table 1 contains the comparison of, I assume, the running time).
> >
> > ***response:*** We apologize for not providing sufficient comparison with baseline methods. In our revised manuscript, we utilized more metrics to evaluate the proposed method in comparison with all baseline models. Specifically, the rmse error of discovered system coefficients, training cost, precision metric $M_P$, recall metric $M_R$ are computed and compared for all methods. Moreover, we have also added one more baseline model [M. Tipping, The relevance vector machine, Advances in neural information processing systems
> > 12 (1999).]) and performed the comparison on two additional test cases, 2-D diffusion equation and 2-D  Poisson equation. Please see the updated Table 1 and more comprehensive results in the updated supplementary materials, Table 3 located in appendix A.7.
> >
> > ### Questions
> >
> > > 1. Equation (3), for PDE, how do you model terms like $u_x$ and $u_{xx}$? Please give an example like Burger's equation. Is the representation in $R^{n\times s\times m}$?
> >
> > ***response:***
> > We thank the reviewer for this question. These terms are approximated by spline bases as
> >     \begin{align*}
> >     u_x(t,x) = \sum_{s_1, s_2} \frac{\partial N_{s_1, s_2}^{k_1, k_2} (t, x)}{\partial x} \theta_{s_1, s_2} = \sum_{s_1, s_2} N_{s_1,k_1}(t)\frac{\partial  N_{s_2, k_2}(x)}{\partial x} \theta_{s_1, s_2}
> > \end{align*}
> > \begin{align*}
> >     u_{xx}(t,x)  = \sum_{s_1, s_2} \frac{\partial^2 N_{s_1, s_2}^{k_1, k_2} (t, x)}{\partial x^2}  \theta_{s_1, s_2} = \sum_{s_1, s_2}  N_{s_1,k_1}(t)\frac{\partial^2  N_{s_2, k_2}(x)}{\partial x^2 }  \theta_{s_1, s_2},
> >     \end{align*}
> >     where are evaluated at collocation points. Following the expression, the number of control points (trainable weights) are represented in $\mathbb{R}^{(r_1+k)\times (r_2+k)}$, where $r_1$,$r_2$ are given empirically, not exceeding $O(10^2)$ in current work. Since $N_{s_i,k_i}(x)$ is non-zero only when $x \in [\tau_{s_i }, \tau_{s_i+1+k_i}] $, therefore, the number of nonzero terms is independent of the $n$, $s$ or $m$. In our implementation, we leverage the sparsity for the computation of $u_x$, $u_{xx}$, $\cdots$. In particular, sparse matrix is used to save the memory. And the dimension/implementation details are provided in Appendix A.9.
> >
> > Admittedly, the tensor product is not the scalable way to extend the spline learning for high-dimensional settings, and there are better ways to achieve so. For example, as shown in Fig. 3 of the paper "SplineCNN: Fast Geometric Deep Learning with Continuous B-Spline Kernels", it would be much better if a continuous B-spline convolutional kernel can be defined and only the basis function in the subdomain is expanded. This would allow a significant reduction of trainable parameters and B-spline basis functions. We found the current result listed in the manuscript is not compromised by our inefficient implementation. These have been explicitly discussed in the revised manuscript.
> >
> > > 2. Equation (10), what is $U$? Same for equation (13).
> >
> > ***response:***
> > We apologize for the confusion. $U$ in Equation (10) should be the solution vector $\mathbf{u}$, and $U$ in equation(13) should be the noisy observation, which is changed to $\hat{\textbf{U}}$. We have updated the notations thoroughly to avoid any confusion in the revised manuscript.

---

> > > ### Author Response · Authors · 2022-08-02
> > > **Response to Reviewer 3: Part 3**
> > >
> > > > 3. Table 1, what are the values?
> > >
> > > ***response:***  The value in original Table 1 is the root mean square error (rmse) of the discovered terms, which is defined in equation (16). To better evaluate the proposed method, two additional metrics, precision $M_P$ and recall $M_R$, are computed in the revision.
> > >
> > > ### Limitations
> > >
> > > > 1. The authors should clarify the limitations of splines in modeling higher-dimension data.
> > >
> > > ***response:***  We thank the reviewer for this suggestion. In the revision, we have updated the discussion section by directly pointing out the limitations of splines in modeling high-dimensional problems and discussing potential improvements in the future

---

> > > > ### Author Response · Authors · 2022-08-02
> > > > **Response to Reviewer 3: Part 4 Additional Table for comparison**
> > > >
> > > > FYI, the metrics are defined as:
> > > >
> > > > $$\textbf{rmse} = \frac{\lVert C_{Discovery}-C_{True}\rVert_{2}}{\lVert C_{True}\rVert_{2}}$$
> > > >
> > > > $$\mathrm{M_P} = \frac{\lVert C_{Discovery}\odot C_{True}\rVert_{0}}{\lVert C_{Discovery}\rVert_{0}}$$
> > > >
> > > > $$\mathrm{M_R} = \frac{\lVert C_{Discovery}\odot C_{True}\rVert_{0}}{\lVert C_{True}\rVert_{0}}$$
> > > >
> > > > where $\odot$ represents element-wise product of vectors and the $l_0$ norm is the non-zero terms in a vector.
> > > >
> > > > | Table: ODE and PDE discovery comparison |  |  |  |  |  |  |
> > > > | :--- | :--- | :--- | :--- | :--- | :--- | :--- |
> > > > | Name | rmse(0%) | rmse(1%) | rmse (large) | $\mathrm{M}_{\mathrm{P}}$ | $\mathrm{M}_{\mathrm{R}}$ | Training Cost |
> > > > |  |  |  | VdP Oscillator |  |  |  |
> > > > | BSL(Ours) | $0.2$ | $2.82$ | $18.04$ | 1 | 1 | $\sim 133(+3) s$ |
> > > > | PINN-SR | Fail | Fail | Fail | $0.214$ | $0.75$ | $\sim 1213 s$ |
> > > > | SINDy | $1.0$ | $1.93$ | Fail | $0.267$ | $1.0$ | $\sim 10 s$ |
> > > > | RVM | $1.0$ | $2.54$ | $27.46$ | 1 | 1 | $\sim 10 s$ |
> > > > |  |  |  | Lorenz 96 |  |  |  |
> > > > | BSL(Ours) | $0.269$ | $1.47$ | $13.0$ | 1 | 1 | $\sim 1654(+438) s$ |
> > > > | PINN-SR | Fail | Fail | Fail | $0.5$ | $0.22$ | $\sim 10788 s$ |
> > > > | SINDy | $0.4$ | $0.64$ | Fail | $0.75$ | 1 | $\sim 10 s$ |
> > > > | RVM | $0.4$ | $0.6$ | $49.7$ | 1 | 1 | $\sim 25 s$ |
> > > > |  |  |  | Advection Equation |  |  |  |
> > > > | BSL(Ours) | $0.26$ | 1 | $1.9$ | 1 | 1 | $\sim 946(+233) s$ |
> > > > | PINN-SR | $5.9$ | $4.5$ | $30.4$ | 1 | 1 | $\sim 650 s$ |
> > > > | SINDy | $2.3$ | $8.2$ | $38.9$ | 1 | 1 | $\sim 10 s$ |
> > > > | RVM | $0.77$ | $6.76$ | Fail | $0.2$ | 1 | $\sim 4 s$ |
> > > > |  |  |  | Burgers' Equation |  |  |  |
> > > > | BSL(Ours) | $3.62$ | $4.13$ | $6.38$ | 1 | 1 | $\sim 117(+74) s$ |
> > > > | PINN-SR | $10.2$ | $3.3$ | $10.3$ | 1 | 1 | $\sim 512 s$ |
> > > > | SINDy | $0.826$ | Fail | Fail | 1 | $0.5$ | $\sim 10 s$ |
> > > > | RVM | $0.754$ | Fail | Fail | $0.1429$ | $0.5$ | $\sim 4 s$ |
> > > > | Name | rmse(0%) | rmse(0.1%) | rmse (large) | $\mathrm{M}_{\mathrm{P}}$ | $\mathrm{M}_{\mathrm{R}}$ | Training Cost |
> > > > |  |  |  | Burgers with Source |  |  |  |
> > > > | BSL(Ours) | 11 | $12.4$ | $13.4$ | 1 | 1 | $\sim 396(+340) s$ |
> > > > | PINN-SR | $10.5$ | 15 | $34.6$ | 1 | 1 | $\sim 600 s$ |
> > > > | SINDy | $26.2$ | Fail | Fail | 1 | $0.33$ | $\sim 10 s$ |
> > > > | RVM | $27.6$ | Fail | Fail | $0.5$ | $0.67$ | $\sim 10 s$ |
> > > > |  |  |  | Heat Equation |  |  |  |
> > > > | BSL(Ours) | 19 | 19 | $38.9$ | 1 | 1 | $\sim 71(+8) s$ |
> > > > | PINN-SR | Fail | Fail | Fail | 0 | 0 | $\sim 285 s$ |
> > > > | SINDy | $1.9$ | 17 | Fail | $0.25$ | 1 | $\sim 10 s$ |
> > > > | RVM | $2.8$ | 6 | Fail | 0 | 0 | $\sim 6 s$ |
> > > > |  |  |  | Poisson Equation |  |  |  |
> > > > | BSL(Ours) | $1.18 \times 10^{-2}$ | $0.133$ | $16.7$ | 1 | 1 | $\sim 92(+9) s$ |
> > > > | PINN-SR | Fail | Fail | Fail | 0 | 0 | $\sim 3737 s$ |
> > > > | SINDy | $1.15$ | 87 | 962 | 1 | 1 | $\sim 10 s$ |
> > > > | RVM | $1.15$ | 232 | 968 | 1 | 1 | $\sim 10 s$ |
> > > > ||||||||

---

> > > > > ### Author Response · Authors · 2022-08-02
> > > > > **Response to Reviewer 3: Part 5 Additional table for implementation**
> > > > >
> > > > > FYI, the number of control points (trainable weights) $\theta$ in 1-d is shown in the second dimension of the matrix, e.g., $54$. And the total number of control points for the 2-d scenario is listed in the last column. We only store the non-zero elements of the two-dimensional basis to leverage the sparsity (local support) of the spline.
> > > > >
> > > > >
> > > > > | Direct tensor-product spline for PDE |  |  |  |  |
> > > > > | :---: | :---: | :---: | :---: | :---: |
> > > > > | basis $x$ | basis $t(y)$ | basis $(x, t(y))$ | sparsity | trainable params |
> > > > > | Advection Equation |  |  |  |  |
> > > > > | $50 \times 54$ | $50 \times 54$ | 62500 | $0.0086$ | 2916 |
> > > > > | Burgers' Equation |  |  |  |  |
> > > > > | $128 \times 13$ | $101 \times 19$ | $92872$ | $0.058$ | $247$ |
> > > > > | Burgers' Equation with Source |  |  |  |  |
> > > > > | $201 \times 103$ | $101 \times 103$ | 251415 | $0.0012$ | $10609$ |
> > > > > | Heat Equation |  |  |  |  |
> > > > > | $51 \times 11$ | $51 \times 11$ | 41209 |0.1309  |121  |
> > > > > | Poisson |  |  |  |  |
> > > > > | $101 \times 53$ | $101 \times 53$ | 162409 |  0.0057|2809  |
> > > > > ||||||

---

> ### Author Response · Authors · 2022-08-08
> **Thank you for your insightful comments**
>
> Dear reviewer nVYU,
>
> Again, we would like to thank your insightful comments on our paper. We believe we have fully addressed your concerns through our response earlier and the revised paper. We do hope to have your feedback and look forward to answering any additional questions you have. Thank you very much.

---

> > ### Comment · Reviewer_nVYU · 2022-08-09
> > **Empirical comparison**
> >
> > Thanks for responding to my questions. I still have concerns about Table 1. First of all, a comparison with SINDy may not be fair. For SINDy,  it would be fair to preprocess the inputs by first smoothing the inputs and then resample new regularly spaced inputs. The table shows that SINDy performs better often times, when it does not fail to fit.

---

> > > ### Author Response · Authors · 2022-08-09
> > > **Response to the concern about the fairness of the comparison with SINDy**
> > >
> > > ***Response:*** We thank the reviewer for this question and concern about the comparison with SINDy. In fact, the settings of SINDy presented in Table 1 (representative cases) and Table 7 (full list of test cases) are optimized based on the open-source repository published by the original authors, where the noisy data has been preprocessed by polynomial smoothing or Tikhonov regularization [1] and uniformly re-sampled afterwards, as the reviewer suggested. We apologize for not reporting these details in the original manuscript, and we will update it by providing all the setting details.
> > >
> > > We acknowledge the significant contribution of SINDy, which is the seminal work of library-based equation discovery algorithms. Almost all the library-based discovery methods, including this work, more or less leverage the idea of sparse regression in SINDy to obtain the most parsimonious model forms.
> > >
> > > SINDy works well for "perfect" data -- as shown in Table 1, in low noise regimes, SINDy performs better some of the time. However, SINDy is known to be less capable of handling corrupted data (sparse and noisy), which is more common in real-world scenarios. The improvement of data smoothing/filtering is limited since the derivatives can be distorted after smoothing, which leads to inaccurate systems with spurious terms.
> > >
> > > To further address the reviewer's concern, we conducted additional experiments for SINDy using different smoothing methods from [1], including smoothing based polynomial interpolation, convolutional smoother, smoothing with Tikhonov regularization and smoothing with spline fitting. Due to the time constrain, we performed comparison of SINDy with these smoothing methods on a representative ODE system (Van der Pol system) and PDE system (Poisson equation) studied in this work. As shown in the Table below, when the data noise is above $5\\%$, although data is preprocessed using smoothing and uniform resampling, none of these method work. Basically, the SINDy still failed to discover the correct model forms with different smoothing schemes, and the identified systems are different from the true. In contrast, our proposed approach is very robust and superior to handling corrupted data, thanks to the spline learning in Bayesian settings.
> > >
> > > ( 'no smoother' means direct finite difference without smoothing, 'poly' means smoothing with polynomial interpolation, 'conv' means convolutional smoother,  'Tikhonov' means smoothing Tikhonov regularization, 'Spline' means spline fitting. More details can be found in [1][2].)
> > >
> > > ***Reference***
> > >
> > > [1]. Rudy, Samuel H., et al. "Data-driven discovery of partial differential equations." Science advances 3.4 (2017): e1602614.
> > >
> > > [2]. de Silva, Brian M., et al. "Pysindy: a python package for the sparse identification of nonlinear dynamics from data." arXiv preprint arXiv:2004.08424 (2020).

---

> > > > ### Author Response · Authors · 2022-08-09
> > > > **Additional Tables for fairness of the comparison with SINDy**
> > > >
> > > > |Smoothing algorithm effects | |
> > > > | :---: | :---: |
> > > > | Name | $\epsilon(Noise) 5\\%$|
> > > > | |Van der Pol Oscillator|
> > > > |True|$\frac{dx}{dt}=y$|
> > > > ||$\frac{dy}{dt} = -x-0.5x^2y+0.5y$|
> > > > |SINDy (No smoother)|$\frac{dx}{dt}=0.0277-0.217x+1.4y-0.025x^2$|
> > > > || $+0.0542x^3-0.1285x^2y+0.1039xy^2-0.0896y^3$|
> > > > ||$\frac{dy}{dt} = -0.9835x+0.3462y-0.4675x^2y+0.0325y^3$|
> > > > | SINDy (Poly)|$\frac{dx}{dt}=0.1197+0.169x+0.9975y-0.0292x^2$ |
> > > > | | $+0.0232xy-0.0274y^2-0.0393x^3+0.0208x^2y-0.0463xy^2$|
> > > > |SINDy (Conv)|$\frac{dx}{dt}=0.1518+0.9978y-0.0486x^2+0.0274xy-0.0327y^2$|
> > > > || $\frac{dy}{dt} = -1.1154x-0.2143y+0.0348x^3-0.4374x^2y+0.0613y^3$|
> > > > |SINDy (Tikhonov)|$\frac{dx}{dt}=-0.2572+0.1486x+1.1299y+0.0982x^2-0.0721xy$|
> > > > ||$+0.0549y^2-0.0478x^2y-0.0816xy^2-0.0542y^3$|
> > > > ||$\frac{dy}{dt} = 0.2338-1.3282x+0.1603y-0.0718x^2$|
> > > > ||$+0.058xy-0.0564y^2+0.1069x^3-0.0842x^2y+0.1373xy^2-0.0415y^3$|
> > > > |SINDy (Spline)|$\frac{dx}{dt}= 0.1916+1.5304y-0.0614x^2+0.0215xy-0.0325y^2-0.155x^2y+0.0796xy^2-0.1168y^3$|
> > > > ||$\frac{dy}{dt} = -0.35-1.0127x+0.6756y+0.0695x^2+0.0741y^2-0.5493x^2y+0.0523xy^2-0.0412y^3$|
> > > > |  | Poisson Equation|
> > > > |True|$u_{yy} = -u_{xx}-sin(x)sin(y)$|
> > > > |SINDy (No smoother)|$u_{yy}=0.5635u_{xx}+1.683uu_{x}+0.088uu_{xx}-0.17sin(x)sin(y)$ |
> > > > |SINDy (Poly)|$u_{yy}=0.2316u_{xx}-0.4221sin(x)sin(y)$|
> > > > |SINDy (Conv)|$u_{yy}=0.47u_{xx}-0.2649sin(x)sin(y)+0.073sin(x)cos(y)$|
> > > > |SINDy (Tikhonov)|$u_{yy}=-0.17u_{xx}-0.087uu_{x}-0.458sin(x)sin(y)+0.02sin(x)cos(y)$|
> > > > ||$+0.058xy-0.0564y^2+0.1069x^3-0.0842x^2y+0.1373xy^2-0.0415y^3$|
> > > > |SINDy (Spline)|$$u_{yy}=0.1562u_{xx}+0.1751uu_{x}-0.3951sin(x)sin(y)-0.0911sin(x)cos(y)$$|
> > > > |||
> > > >
> > > > | Table: ODE and PDE discovery comparison using different smoothing methods|  |  |  |  |
> > > > | :---  | :--- | :--- | :--- | :--- |
> > > > | Name | rmse (large) | $\mathrm{M}_{\mathrm{P}}$ | $\mathrm{M}_{\mathrm{R}}$ | Training Cost |
> > > >  |  | Van der Pol Oscillator|  |  |  |
> > > > | BSL (Ours)  | $18.04$ | 1 | 1 | $\sim 133(+3)s$ |
> > > > | SINDy (No smoother)  | Fail | $0.33$ | $1$ | $\sim 10 s$ |
> > > > | SINDy (Poly)  | Fail | $0.6$ | $0.75$ | $\sim 10 s$ |
> > > > | SINDy (Conv)  | Fail | 0.4 | 1 | $\sim 10 s$ |
> > > > |SINDy (Tikhonov)|Fail|0.21|1|$\sim 10s$|
> > > > |SINDy (Spline)|Fail|0.25|1|$\sim 10s$|
> > > >  |  | Poisson Equation|  |  |  |
> > > > | BSL(Ours)  | $16.7$ | 1 | 1 | $\sim 92(+9)s$ |
> > > > | SINDy (No smoother)  | $\text{Fail}$ | 0.5 | 1 | $\sim 10s$ |
> > > > | SINDy (Poly)  | $962$ | 1 | 1 | $\sim 10s$ |
> > > > | SINDy (Conv)  | $\text{Fail}$ | 0.66 | 1 | $\sim 10s$ |
> > > > | SINDy (Tikhonov)  | $\text{Fail}$ | 0.5 | 1 | $\sim 10s$ |
> > > > | SINDy (Spline)  | $\text{Fail}$ | 0.5 | 1 | $\sim 10s$ |
> > > > ||||||

---

> > > > > ### Comment · Reviewer_nVYU · 2022-08-09
> > > > > **Please publish reproducible experiment code**
> > > > >
> > > > > Thanks for the additional experiments and the improved manuscript. For reproducibility, it is important that the experiments are described in detail and the code published. I have updated my score.

---

> > > > > > ### Author Response · Authors · 2022-08-10
> > > > > > **Thank you so much**
> > > > > >
> > > > > > Thank you so much for all the constructive discussions and for raising our score to 6! We will for sure include more details and publish the code for reproducibility.

---

### Official Review · Reviewer_7qjr · 2022-07-15

**Rating:** 7
**Confidence:** 2
**Soundness:** 3 good
**Presentation:** 2 fair
**Contribution:** 3 good

**Summary:**

In this paper, the authors:
- propose a system identification algorithm that leverages spline bases to fit sparsely fit noisy data, and provides parameter and prediction uncertainty (which can be used in an active learning / Bayesian optimization setting)
- show that the algorithm can be used to infer ODEs and PDEs, and with high level of noise
- show that the proposed algorithm performs well against two other standard / state of the art methods, but has tradeoffs at different noise levels

**Questions:**

- in Figure 2, the discovered mean of the library terms are plotted against their actual values: what is the actual value in these cases for the VdP and Lorenz96 systems? If I understand correctly, the output of the algorithm are the weights of the splines at different orders, which do not relate to the actual dynamical equations for the systems at all, so how are the comparisons made? Or do the library terms CX2 and CX3 correspond to, e.g., first and second order terms such as x, and xy? It would be nice if this point could be clarified in the text

- in this vein, and it's very possible that I'm confused on this point, but the motivation was to find interpretable models that match the true system. For example, with SINDY, the library can be constructed with polynomial terms up to arbitrary orders, which can then be exactly matched with the original system. In this work, the basis functions are splines, which do not allow for such interpretations

- the schematic in Figure 1 demonstrates a nice effort in explaining the algorithm, but it's a bit crowded, and it's especially unclear how the loss functions in the red box relate to each other, and what the correspond to in the main? In particular, it's a bit unclear to me how the data fitting loss (L2) is combined with the logL loss?

- related to my confusion on the learning procedure, are the spline parameters separately learned (as indicated by the blue box being separated)? or is the whole thing trained end to end. I apologize for the confusion here and do not really have concrete suggestions for how to improve clarity.

- while I understand that the algorithm comparisons are conducted on the benchmark ODE/PDEs, it would be nice to see how the proposed method works with real noisy data that is routinely modeled with one of the systems (e.g., advection equation), or another i.e. predator-prey data with LotkaVolterra, or COVID numbers with SIR model

- figure fonts are extremely small, basically unreadable in many of them

- some terminology should be explained for a reader unfamiliar with SI, e.g., "control points"

**Ethics Review Area:**

["I don’t know"]

**Limitations:**

a brief discussion on library choice is included, but a longer discussion is warranted regarding the splines, as well as how it performs in high noise regimes (especially in comparison with the other algorithms). Furthermore, potential negative societal impact should be discussed as such an algorithm can have broad consequences in accurately modeling natural systems.

**Strengths And Weaknesses:**

originality: I don't know the literature well enough to know if others have attempted similar approaches, but in comparison to standard SINDY, this work builds upon the idea of constructing the system from sparse basis functions but additionally provides a probabilistic output

quality: I did not check the math/code, but I believe the results are of high quality at a superficial level, and support the overall claim (i.e., uses splines to discovery system equations, is sparse, and provides uncertainty estimates). Unclear if it would work beyond the standard benchmark synthetic systems.

clarity: the description of the algorithm is a bit unclear (specific issues outlined below), but the overall goal, the components of the algorithm, and the results were clearly communicated

significance: system identification that tries to retrieve the underlying mechanistic model is especially important in the context of scientific applications with noisy observations; has potential to be broadly impactful in many areas of ML and science

---

> ### Author Response · Authors · 2022-08-02
> **Response to Reviewer 2: Part 1**
>
> ### Strengths And Weaknesses:
> ***Response:*** We sincerely thank the reviewer's overall assessment and appreciate his/her constructive comments and suggestions, which are very helpful for improving our paper.
>
> In the existing literature, the state-of-the-art data-driven interpretable learning methods mainly use either finite difference (FD) or auto differentiation (AD) to compute derivatives, which is critical for explicit model form discovery. However, the FD is very sensitive to data noise, while the AD approach generally does not work for ODE systems. This work bridges the gap and can simultaneously identify both ODE and PDE systems from corrupted datasets, meanwhile, the associated uncertainty is well quantified without introducing much computational overhead. We believe this work is an important contribution to the literature in this area.
>
> We here respond in detail to reviewer’s comments and highlight appropriate changes to the revised manuscript.
>
>
> ### Questions
> > 1. in Figure 2, the discovered mean of the library terms are plotted against their actual values: what is the actual value in these cases for the VdP and Lorenz96 systems? If I understand correctly, the output of the algorithm are the weights of the splines at different orders, which do not relate to the actual dynamical equations for the systems at all, so how are the comparisons made? Or do the library terms CX2 and CX3 correspond to, e.g., first and second order terms such as x, and xy? It would be nice if this point could be clarified in the text.
>
> ***Response:*** We thank the reviewer for the question, which has not been clarified in the original manuscript. We here explain it in detail and will revise the manuscript accordingly.
>
> 1. The ``True Coefficient" in Figure 2 represents true values of equation coefficients of the system to be identified. For example, the true governing equation for VdP is $\frac{dx}{dt} = y$ and $\frac{dy}{dt} = -x-0.5x^2y+0.5y$. The system equation coefficients are denoted as $\{x^{(2)},y^{(1)},y^{(2)},x^2y^{(2)}\}$ with the true value $\{x^{(2)} = -1, y^{(1)} = 1, y^{(2)} = 0.5, x^2y^{(2)} = -0.5\}$, where the superscript represent the corresponding equations. Similarly, for the Lorenz 96 system, the compact form of the governing equations can be written as $\frac{dX_i}{dt} = (X_{i+1}-X_{i-2})X_{i-1}-X_i+F$, with periodic boundary conditions $X_{-1}=X_{n-1}$, $X_{0}=X_{n}$ and $X_{n+1}=X_1$. When $n = 6$ it can be expanded as $\frac{dX_1}{dt} = (X_2-X_5)X_6-X_1+F$, $\frac{dX_2}{dt} = (X_3-X_6)X_1-X_2+F$, $\frac{dX_3}{dt} = (X_4-X_1)X_2-X_3+F$, $\frac{dX_4}{dt} = (X_5-X_2)X_3-X_4+F$, $\frac{dX_5}{dt} = (X_6-X_3)X_4-X_5+F$, $\frac{dX_6}{dt} = (X_1-X_4)X_5-X_6+F$. The system coefficients are represented by $\{F^{(1)},F^{(2)},F^{(3)},F^{(4)},F^{(5)},F^{(6)},X_1^{(1)}, X_2^{(2)},X_3^{(3)},X_4^{(4)}, X_5^{(5)}, X_6^{(6)}, X_1X_2^{(3)}, X_1X_3^{(2)},\\ X_1X_5^{(6)}, X_1X_6^{(2)}, X_2X_3^{(4)}, X_2X_4^{(3)}, X_2X_6^{(1)}, X_3X4^{(5)}, X_3X_5^{4}, X_4X_5^{(6)}, X_4X_6^{(5)}, X_5X_6^{(1)}\}$ with the true values of $\{8,8,8,8,8,8,-1,-1,-1,-1,-1,-1,-1,1,1,-1,-1,1,1,-1,1,-1,1,-1\}$. More details are provided in the caption and supplementary materials in the revised version.
>
> 2. Although the outputs of the BSL algorithm are the weights of the splines at different orders, these spline weights will be used to reconstruct the library terms based on spline basis. For example, as $\mathbf{x} = \mathbf{N}\mathbf{\theta}$, where $\mathbf{N}$ is the spline basis and $\mathbf{\theta}$ is its weights, we can easily obtain $\frac{d\mathbf{x}}{d\mathbf{t}} = \mathbf{\frac{dN}{dt}}\mathbf{\theta}$, where $\mathbf{\frac{dN}{dt}}$ are the analytical derivatives of the spline basis. Namely, all the library terms can be reconstructed by spline basis functions (and its $k^\mathrm{th}$ derivatives) and spline weights. So the reviewer is correct, the library terms $CX2$ and $CX3$ correspond to first and second order polynomial terms in ODE discovery, which is constructed based on the trained splines. For better readability, we already updated Fig.2 and replace $CX2,CX3...$ by the actual library terms.

---

> > ### Author Response · Authors · 2022-08-02
> > **Response to Reviewer 2: Part 2**
> >
> > > 2. in this vein, and it's very possible that I'm confused on this point, but the motivation was to find interpretable models that match the true system. For example, with SINDY, the library can be constructed with polynomial terms up to arbitrary orders, which can then be exactly matched with the original system. In this work, the basis functions are splines, which do not allow for such interpretations
> >
> > ***Response:*** We thank the reviewer for this question, which we didn't make clear in the original manuscript. There are two levels of basis functions in this work: one is the functional basis (i.e., candidate terms) to build the library, and the other is the spline basis to construct the candidate terms. In SINDy, the library is constructed based on polynomials of state variables $x, y$, which works fine when data is sufficient and noise-free. However, the SINDy family fails to deal with the data with large noise and sparsity. In this work, spline basis is leveraged to tackle these challenges, where the states (response surfaces) are reconstructed by learnable weights combined with spline basis. The spline basis is not directly used as the basis functions to build the library, instead, the library is constructed hierarchically and can include terms up to polynomial terms in arbitrary orders, just like SINDy. For example, in ODE discovery, the spline basis is used to reconstruct the monomials and then the monomials are used to construct the polynomials. The entire learning process is end-to-end, as described by Equations (8) and (15). The optimal values of spline control points (trainable weights) $\mathbf{\theta}$ and functional basis coefficients $\mathbf{W}$ are learned simultaneously, and once the optimal values of $\mathbf{W}$ are obtained, the underlying system is identified and its model forms can be interpreted explicitly.
> >
> > > 3. the schematic in Figure 1 demonstrates a nice effort in explaining the algorithm, but it's a bit crowded, and it's especially unclear how the loss functions in the red box relate to each other, and what the correspond to in the main? In particular, it's a bit unclear to me how the data fitting loss (L2) is combined with the logL loss?
> >
> > ***Response:***
> > We apologize for the unclear and crowded schematic figure, which will be updated in the revised version to better illustrate the whole algorithm. Back to the question about the training process, it consists of two parts: (1) spline learning and (2) Bayesian alternative direction optimization (ADO). For spline learning, the goal is to fit the state response surfaces using spline basis, and the spline weights are optimized deterministically based on the data-fitting loss. The Bayesian ADO training can be divided into two stages. In stage 1, both the spline weights $\mathbf{\theta}$ and equation term coefficients $\mathbf{W}$ are updated simultaneously, based on the probabilistic loss function (i.e., log likelihood function logL). In stage 2, only the equation term coefficients $ \mathbf{W}_{n \times d} $   are updated by sparsity-promoting regression to further prune out redundant terms.
> >
> > This generates a reduced matrix $ \mathbf{W}_{m \times d} $, where $m \leq n$. Stages 1 and 2 will be conducted iteratively, which is illustrated by the pink circular arrows in Fig. 1.
> >
> > > 4. related to my confusion on the learning procedure, are the spline parameters separately learned (as indicated by the blue box being separated)? or is the whole thing trained end to end. I apologize for the confusion here and do not really have concrete suggestions for how to improve clarity.
> >
> > ***Response:*** We thank the reviewer for this question. Yes, the entire learning process is end-to end. Namely, the spline learning is performed within the Bayesian ADO framework and all the parameters are optimized simultaneously. In the schematic figure, however, we try to separate the whole process to three components for illustrative purposes.
> >
> > > 5. while I understand that the algorithm comparisons are conducted on the benchmark ODE/PDEs, it would be nice to see how the proposed method works with real noisy data that is routinely modeled with one of the systems (e.g., advection equation), or another i.e. predator-prey data with LotkaVolterra, or COVID numbers with SIR model
> >
> > ***Response:***
> > We thank the reviewer for this constructive suggestion, which will surely improve the quality of this manuscript. We plan to implement more realistic cases as the reviewer suggested, but might not be able to finish the new testing case due to the limited time window for revision. However, we will try our best to get this done during the author-review discussion period and update the manuscript accordingly.
> >
> > > 6. figure fonts are extremely small, basically unreadable in many of them
> >
> > ***Response:***
> > We thank the reviewer for pointing out this issue. We have enlarged the font sizes and made them readable in the revised manuscript.

---

> > > ### Author Response · Authors · 2022-08-02
> > > **Response to Reviewer 2: Part 3 Additional Tables**
> > >
> > > > 7. some terminology should be explained for a reader unfamiliar with SI, e.g., "control points"
> > >
> > > ***Response:***
> > > We thank the reviewer's suggestions. We have added explanations for the undefined terminology either in the main text or in the supplementary materials due to the space limit.
> > >
> > > | Terminologies | |
> > > | :---: | :---: |
> > > |Name |Explanation|
> > > | Control points| Trainable weight $\theta$ for spline basis. |
> > > |  Measurement points|Sparse spatio-temporal points with labels.|
> > > | Collocation points|Dense spatio-temporal points without labels.|
> > > | Library candidates| A collection of polynomial terms that|
> > > ||the system identification algorithm can choose parsimonious terms from it|
> > > || e.g., $\{x,y,x^2y,...\}$ (for ODE) or $\{u,uu_x,u_{xx}...\}$ for (PDE). |
> > > | ADO iteration|Alternating direction optimization to update the trainable parameters, |
> > > ||including control points $\mathbf{\theta}$, weight of library candidates $\mathbf{W}$ and covariance matrices.|
> > > | Aleatoric Uncertainty|Due to intrinsic randomness by nature, which is irreducible. |
> > > |Epistemic Uncertainty| Because of a lack of knowledge, which can be reduced by adding more information.
> > > |||
> > >
> > >
> > >
> > > ### Limitations
> > >
> > > > 1.a brief discussion on library choice is included, but a longer discussion is warranted regarding the splines, as well as how it performs in high noise regimes (especially in comparison with the other algorithms). Furthermore, potential negative societal impact should be discussed as such an algorithm can have broad consequences in accurately modeling natural systems.
> > >
> > > ***Response:***
> > > We thank the reviewer for the valuable suggestions. We further discuss why the proposed method based on splines outpefrom other algorithms in high-noise regimes. Moreover, following the reviewer's suggestion, we also discussed potential negative social impacts if it is abusively used to model natural systems. More details can be found in the revised discussion section.

---

> > > > ### Author Response · Authors · 2022-08-04
> > > > **Response to Reviewer 2: Part 4 Predator-prey data with LotkaVolterra**
> > > >
> > > > > 5. while I understand that the algorithm comparisons are conducted on the benchmark ODE/PDEs, it would be nice to see how the proposed method works with real noisy data that is routinely modeled with one of the systems (e.g., advection equation), or another i.e. predator-prey data with LotkaVolterra, or COVID numbers with SIR model
> > > >
> > > > ***Response:***
> > > > Thank you for the reviewer's constructive suggestion. We have made one more real case, Predator-prey data with LotkaVolterra, and the supplementary PDF is also updated to include this case. We will also give more details about the case setting and the experiment's result.
> > > >
> > > > The real data set is obtained online and it depicts the population of hares and lynx from 1900 to 1920 from Hudson Bay Company.
> > > > The real data is attached in the end. The reference governing equation by mathematical analysis is:
> > > >
> > > > $\frac{dx}{dt} = 0.4807x-0.0248xy$
> > > >
> > > >  $\frac{dy}{dt} =-0.9272y+0.0276xy$
> > > >
> > > >
> > > >  We test the 4 methods on this data set and the result can be found in the Table below:
> > > >
> > > > FYI, the metrics are defined as:
> > > >
> > > > $$\textbf{rmse} = \frac{\lVert C_{Discovery}-C_{True}\rVert_{2}}{\lVert C_{True}\rVert_{2}}$$
> > > >
> > > > $$\mathrm{M_P} = \frac{\lVert C_{Discovery}\odot C_{True}\rVert_{0}}{\lVert C_{Discovery}\rVert_{0}}$$
> > > >
> > > > $$\mathrm{M_R} = \frac{\lVert C_{Discovery}\odot C_{True}\rVert_{0}}{\lVert C_{True}\rVert_{0}}$$
> > > >
> > > > where $\odot$ represents element-wise product of vectors and the $l_0$ norm is the non-zero terms in a vector.
> > > >
> > > > | Table: ODE discovery comparison for predator-prey system |  |  |  |  |
> > > > | :---  | :--- | :--- | :--- | :--- |
> > > > | Name | rmse (large) | $\mathrm{M}_{\mathrm{P}}$ | $\mathrm{M}_{\mathrm{R}}$ | Training Cost |
> > > >  |  | Predator-prey|  |  |  |
> > > > | BSL(Ours)  | $30.4$ | 1 | 1 | $\sim 2278(+6) s$ |
> > > > | PINN-SR | Fail | $0.5$ | $0.25$ | $\sim 2988 s$ |
> > > > | SINDy | Fail | $0.6$ | $0.75$ | $\sim 10 s$ |
> > > > | RVM  | Fail | 0.8 | 1 | $\sim 10 s$ |
> > > > ||||||
> > > >
> > > > We have made assumptions about constructing the libraries. We assume the predator (lynx) only feeds on the prey (hares). Meantime, the prey (hares) only has one predator (lynx). Therefore, the change rate of these two species can only depend on themselves ($x$,$y$) and some higher order correlations between them ($xy,x^2y,xy^2$). The discovered forms of the 4 methods are listed in the end. Finally, the UQ results are shown in Fig.10 in appendix A.11. In short, only the proposed method works on the real sparse and noisy dataset. And the UQ prediction covers more measurement points than the reference model, which helps to explain the data set better.
> > > >
> > > > |Analytical ODE | |
> > > > | :---: | :---: |
> > > > | Name | $\epsilon$ (large) |
> > > > |  | Predator-prey (Lotka-Volterra) |
> > > > | True | $\frac{dx}{dt}=0.4807x-0.0248xy$, $\frac{dy}{dt} = -0.9272y+0.0276xy$ |
> > > > | BSL(Ours) | $\frac{dx}{dt}=-0.5124(\pm0.028)x-0.0266(\pm8.72\times10^{-4})xy$, $\frac{dy}{dt} =-0.9258(\pm0.065)y+0.0279(\pm1.47\times10^{-3})xy $ |
> > > > | PINN-SR | $\frac{dx}{dt} = -13.9238y$|
> > > > | |$\frac{dy}{dt}=-0.1144y$|
> > > > |SINDy |$\frac{dx}{dt}=0.5813x-0.0261xy$|
> > > > ||$\frac{dy}{dt} = 0.2549x-0.2702y$|
> > > > |RVM|$\frac{dx}{dt}=0.5732(\pm0.6488)x-0.2386(\pm0.4643)y-0.0253(\pm0.1432)xy$|
> > > > ||$\frac{dy}{dt} =-0.8018(\pm0.9459)y+0.0226(\pm0.1481)xy $|
> > > > |||
> > > >
> > > > |Lynx-Hares population | ||
> > > > | :---: | :---: |:---: |
> > > > | Year | Hares($\times 1000$) |Lynx($\times 1000$)|
> > > > |1900 |30 |4 |
> > > > |1901 |47.2 |6.1 |
> > > > |1902 | 70.2| 9.8|
> > > > |1903 | 77.4| 35.2|
> > > > |1904 |36.3 | 59.4|
> > > > |1905 | 20.6|41.7 |
> > > > |1906 |18.1 |19 |
> > > > |1907 |21.4 | 13|
> > > > |1908 |22 | 8.3|
> > > > |1909 |25.4 |9.1 |
> > > > |1910 |27.1 | 7.4|
> > > > |1911 |40.3 | 8|
> > > > | 1912|57 |12.3 |
> > > > |1913 | 76.6|19.5 |
> > > > |1914 | 52.3|45.7 |
> > > > |1915 | 19.5| 51.1|
> > > > |1916 |11.2 | 29.7|
> > > > | 1917|7.6 | 15.8|
> > > > |1918 |14.6 | 9.7|
> > > > | 1919|16.2 | 10.1|
> > > > |1920 |24.7 |8. 6|
> > > > ||||

---

> > > > > ### Comment · Reviewer_7qjr · 2022-08-08
> > > > > **thanks for the response**
> > > > >
> > > > > extensive and detailed response from the authors, including improved presentation and new experiments on real-world data, which addressed most of my confusions and concerns. best of luck!

---

> > > > > > ### Author Response · Authors · 2022-08-08
> > > > > > **Thank you!**
> > > > > >
> > > > > > Thank you so much for giving us feedback and raising our scores!

---

### Official Review · Reviewer_R79g · 2022-07-25

**Rating:** 5
**Confidence:** 2
**Soundness:** 2 fair
**Presentation:** 2 fair
**Contribution:** 2 fair

**Summary:**

The paper proposes a Bayesian spline learning framework to discover Nonlinear dynamics (ODE or PDE) using sparse data, which is interpretable and is able to quantify the uncertainty. Technical proposals are made to implement this strategy.

**Questions:**

I have several questions regarding clarifying the contribution of the proposed method.

1. In 3.3, the authors assume that Aleatoric Uncertainty is $\epsilon_1 \sim N(0,B)$ and Epistemic Uncertainty is $\epsilon_2 \sim N(0,P)$. Since two noises are independent, the sum of two noises is $N(0, P+B)$, which can be understood as Aleatoric Uncertainty only. Can you elaborate how the proposed methodology differentiate between Aleatoric Uncertainty $N(0,B)$ and Epistemic Uncertainty $N(0,P)$ and how it affects the result?

2. It looks like that the authors try to prove the spline fitting error bound with respect to the mean, which is not zero. Have you considered using this upper bound as a means of the Epistemic Uncertainty? Or have you considered other distributions(e.g. uniform distribution) for $\epsilon_2$ (Epistemic Uncertainty)?

3. Can you please provide additional details on how long it takes to train the model? The proposed method in the paper uses the posterior distribution to estimate uncertainty. However, the key challenge in the Bayesian Inference is estimating posterior distribution. I’m curious to see how much computation you need in order to make the result better.


**Limitations:**

1. The authors recognize that the proposed method relies on a predefined library of candidate terms

2. There are some papers considering finding the best knot size $(\tau_{s+1} - \tau_{s})$ as a way to improve the expressibility of spline fitting function, for instance, "Neural Spline Flow", Conor Durkan, Artur Bekasov, Iain Murray and George Papamakarios. I think this method could potentially reduce the total computation required.


**Strengths And Weaknesses:**

The strengths of the paper are listed below:
1. The experiment results seem promising.
2. The proposed methodology is interpretable and is able to quantify the uncertainty.

The weakness of the paper are listed below:
1. The suggested method relies on the predefined library.
2. The authors did not include the training time to compare the result.

Applying spline based learning with alternating direction optimization to train Nonlinear Dynamics are original, significant. However, I have further questions about the assumptions and results.

---

> ### Author Response · Authors · 2022-08-02
> **Response to Reviewer 1: Part 1**
>
> We sincerely thank the reviewer for the insightful comments and suggestions, which helped to improve the manuscript. We here respond in detail to each comment/question and highlight appropriate changes to the manuscript.
>
> ### Strengths And Weaknesses:
> > 1. The suggested method relies on the predefined library.
>
> ***Response:*** The reviewer is correct. One limitation of the proposed method is that it still relies on a predefined library, which is rooted in prior knowledge of first principles. In general, this is a common limitation for most data-driven equation discovery methods, and it will be very challenging to discover model forms without any prior knowledge.
>
> > 2. The authors did not include the training time to compare the result.
>
> ***Response:*** We apologize for not reporting the training time. In the revised manuscript, we have updated Table 1 by computing more informative metrics and statistics, including training cost, precision $M_p$ and recall $M_R$ and 1 more baseline model. Due to the space limitation, we put a more comprehensive Table in appendix, which is located at Appendix A.7, Table 3.
>
> ### Questions:
> > 1. In 3.3, the authors assume that Aleatoric Uncertainty is $\epsilon_1 \sim \mathcal{N}(0,B)$ and Epistemic Uncertainty is $\epsilon_2\sim \mathcal{N}(0,P)$. Since two noises are independent, the sum of two noises is $\mathcal{N}(0,P+B)$, which can be understood as Aleatoric Uncertainty only. Can you elaborate on how the proposed methodology differentiates between Aleatoric Uncertainty $\mathcal{N}(0,B)$ and Epistemic Uncertainty $\mathcal{N}(0,P)$ and how it affects the result?
>
> ***Response:*** We thank the reviewer for the question. In the Bayesian learning framework, uncertainty can be categorized into two types: aleatoric uncertainty and epistemic uncertainty, the natures of which are very different. Aleatoric uncertainty is due to intrinsic randomness by nature, which is irreducible, while epistemic uncertainty is because of a lack of knowledge, which can be reduced by adding more information. To be specific, in this framework, $\epsilon_1$ represents data uncertainty due to measurement noise, and $\epsilon_2$ represents model-form uncertainty due to data sparsity and model assumptions (e.g., inadequacy of the predefined library). Although, mathematically, the two types of uncertainty can be combined as $\mathcal{N}(0,P+B)$ because of the Gaussian assumption, it is beneficial to quantify these two separately. First, they are physically different by nature and associated with two different equations, the observation equation and forward evolution equation; second, the two types of uncertainty cannot be easily combined for non-Gaussian cases; moreover, in order to perform data assimilation (e.g., EnKF) for better online prediction, the information of both data noise and model-form uncertainty is required. The estimation of $\epsilon_2$ is required for forward propagation of the evolution equation, whereas assimilating new observations requires the propagated $\epsilon_2$ combined with $\epsilon_1$.
>
>
> > 2. It looks like that the authors try to prove the spline fitting error bound with respect to the mean, which is not zero. Have you considered using this upper bound as a means of the Epistemic Uncertainty? Or have you considered other distributions (e.g. uniform distribution) for $\epsilon_2$ (Epistemic Uncertainty)?
>
> ***Response:*** We thank the reviewer for the question. In appendix A.2, we study the spline representation error, which consists of two parts: the first part is due to spline interpolation error, and the second part is due to observation-induced spline fitting error. The error bounds are based on observation errors (aleatoric uncertainty). So far, we've only used error bounds to understand the universal approximation and consistency of spline learning.
>
> It is actually a very good point to use the upper bound to study the epistemic uncertainty or study how this fitting error affects the epistemic uncertainty estimation. However, due to the complication of the this algorithm, which also include further sparse regularization and Bayesian estimation, the answer is not clear to us at this moment. We will further explore it theoretically with new proofs methods in our future work.
>
> In this work, only the Gaussian is used, and we have not considered other distributions yet. However, the proposed method can be easily used for other non-Gaussian distributions. Assume the density function of $\epsilon_2$ is $\rho$, which is parameterized by $P$, then the likelihood equation (15) becomes $$p(\dot{\mathbf{U}}|\mathbf{W},\mathbf{P},\boldsymbol{\theta}) \propto \rho( \dot{N}_c\boldsymbol{\theta} - \boldsymbol{\Phi}(N_c\boldsymbol{\theta})\boldsymbol{W})$$

---

> > ### Author Response · Authors · 2022-08-02
> > **Response to Reviewer 1: Part 2**
> >
> > > 3. Can you please provide additional details on how long it takes to train the model? The proposed method in the paper uses the posterior distribution to estimate uncertainty. However, the key challenge in the Bayesian Inference is estimating posterior distribution. I’m curious to see how much computation you need in order to make the result better.
> >
> > ***Response:*** We thank the reviewer's suggestion, and we have provided details on the training cost of the proposed method. As the reviewer mentioned, quantifying uncertainty in a Bayesian setting will increase the training cost compared to the deterministic setting. However, in this work, we adopted an approximate Bayesian learning approach, Stochastic Weight Averaging-Gaussian (SWAG) method [W. J. Maddox, P. Izmailov, T. Garipov, D. P. Vetrov, A. G. Wilson, A simple baseline for
> > bayesian uncertainty in deep learning, Advances in Neural Information Processing Systems 32
> > (2019).], which is much more efficient and scalable than classic Bayesian approaches, e.g., Markov Chain Monte Carlo (MCMC) sampling. As shown in the table below, we provided training costs of spline learning in both deterministic and Bayesian settings, where the computational overhead due to Bayesian UQ is marked as (+X) s. The training cost of the proposed method is higher than SINDy, which only relies on sparse regression, but the cost is much lower than PINN-SR thanks to the use of spline basis to reduce the search space. The overhead of Bayesian formulation is marginal for all cases.
> >
> > ### Limitations:
> >
> > > 1.The authors recognize that the proposed method relies on a predefined library of candidate terms
> >
> > ***Response:*** The reviewer is correct.  See weakness 1.
> >
> > > 2. There are some papers considering finding the best knot size $(\tau_{s+1}-\tau_s)$ as a way to improve the expressibility of spline fitting function, for instance, "Neural Spline Flow", Conor Durkan, Artur Bekasov, Iain Murray and George Papamakarios. I think this method could potentially reduce the total computation required.
> >
> > ***Response:*** We thank the reviewer for this constructive suggestion, which will significantly improve the proposed method in terms of computation cost for high-dimensional problems. Actually, the proposed spline learning method based on tensor product is less scalable to high-dimensional problems. The ``Neural Spline Flow" [1] mentioned by the reviewer has very good potential to tackle this challenge when moving forward to high-dimensional PDE discovery. Moreover, we have further reviewed recent works in this area and found several novel ways to deal with memory and computation issues (like using a continuous spline convolutional kernel [2]), which will be explored in our future study. The limitations of the current method and potential extensions have been discussed in the revised manuscript.
> >
> > ***Reference***
> >
> > [1]. C. Durkan, A. Bekasov, I. Murray, G. Papamakarios, Neural spline flows, Advances in neural information processing systems 32 (2019)
> >
> > [2]. M. Fey, J. E. Lenssen, F. Weichert, H. Müller, Splinecnn: Fast geometric deep learning with
> > continuous b-spline kernels, Proceedings of the IEEE Conference on Computer Vision and
> > Pattern Recognition, 2018, pp. 869–877.

---

> > > ### Author Response · Authors · 2022-08-02
> > > **Response to Reviewer 1: Part 3 Additional Tables**
> > >
> > > FYI, the metrics are defined as:
> > >
> > > $$\textbf{rmse} = \frac{\lVert C_{Discovery}-C_{True}\rVert_{2}}{\lVert C_{True}\rVert_{2}}$$
> > >
> > > $$\mathrm{M_P} = \frac{\lVert C_{Discovery}\odot C_{True}\rVert_{0}}{\lVert C_{Discovery}\rVert_{0}}$$
> > >
> > > $$\mathrm{M_R} = \frac{\lVert C_{Discovery}\odot C_{True}\rVert_{0}}{\lVert C_{True}\rVert_{0}}$$
> > >
> > > where $\odot$ represents element-wise product of vectors and the $l_0$ norm is the non-zero terms in a vector.
> > >
> > > | Table: ODE and PDE discovery comparison |  |  |  |  |  |  |
> > > | :--- | :--- | :--- | :--- | :--- | :--- | :--- |
> > > | Name | rmse(0%) | rmse(1%) | rmse (large) | $\mathrm{M}_{\mathrm{P}}$ | $\mathrm{M}_{\mathrm{R}}$ | Training Cost |
> > > |  |  |  | VdP Oscillator |  |  |  |
> > > | BSL(Ours) | $0.2$ | $2.82$ | $18.04$ | 1 | 1 | $\sim 133(+3) s$ |
> > > | PINN-SR | Fail | Fail | Fail | $0.214$ | $0.75$ | $\sim 1213 s$ |
> > > | SINDy | $1.0$ | $1.93$ | Fail | $0.267$ | $1.0$ | $\sim 10 s$ |
> > > | RVM | $1.0$ | $2.54$ | $27.46$ | 1 | 1 | $\sim 10 s$ |
> > > |  |  |  | Lorenz 96 |  |  |  |
> > > | BSL(Ours) | $0.269$ | $1.47$ | $13.0$ | 1 | 1 | $\sim 1654(+438) s$ |
> > > | PINN-SR | Fail | Fail | Fail | $0.5$ | $0.22$ | $\sim 10788 s$ |
> > > | SINDy | $0.4$ | $0.64$ | Fail | $0.75$ | 1 | $\sim 10 s$ |
> > > | RVM | $0.4$ | $0.6$ | $49.7$ | 1 | 1 | $\sim 25 s$ |
> > > |  |  |  | Advection Equation |  |  |  |
> > > | BSL(Ours) | $0.26$ | 1 | $1.9$ | 1 | 1 | $\sim 946(+233) s$ |
> > > | PINN-SR | $5.9$ | $4.5$ | $30.4$ | 1 | 1 | $\sim 650 s$ |
> > > | SINDy | $2.3$ | $8.2$ | $38.9$ | 1 | 1 | $\sim 10 s$ |
> > > | RVM | $0.77$ | $6.76$ | Fail | $0.2$ | 1 | $\sim 4 s$ |
> > > |  |  |  | Burgers' Equation |  |  |  |
> > > | BSL(Ours) | $3.62$ | $4.13$ | $6.38$ | 1 | 1 | $\sim 117(+74) s$ |
> > > | PINN-SR | $10.2$ | $3.3$ | $10.3$ | 1 | 1 | $\sim 512 s$ |
> > > | SINDy | $0.826$ | Fail | Fail | 1 | $0.5$ | $\sim 10 s$ |
> > > | RVM | $0.754$ | Fail | Fail | $0.1429$ | $0.5$ | $\sim 4 s$ |
> > > | Name | rmse(0%) | rmse(0.1%) | rmse (large) | $\mathrm{M}_{\mathrm{P}}$ | $\mathrm{M}_{\mathrm{R}}$ | Training Cost |
> > > |  |  |  | Burgers with Source |  |  |  |
> > > | BSL(Ours) | 11 | $12.4$ | $13.4$ | 1 | 1 | $\sim 396(+340) s$ |
> > > | PINN-SR | $10.5$ | 15 | $34.6$ | 1 | 1 | $\sim 600 s$ |
> > > | SINDy | $26.2$ | Fail | Fail | 1 | $0.33$ | $\sim 10 s$ |
> > > | RVM | $27.6$ | Fail | Fail | $0.5$ | $0.67$ | $\sim 10 s$ |
> > > |  |  |  | Heat Equation |  |  |  |
> > > | BSL(Ours) | 19 | 19 | $38.9$ | 1 | 1 | $\sim 71(+8) s$ |
> > > | PINN-SR | Fail | Fail | Fail | 0 | 0 | $\sim 285 s$ |
> > > | SINDy | $1.9$ | 17 | Fail | $0.25$ | 1 | $\sim 10 s$ |
> > > | RVM | $2.8$ | 6 | Fail | 0 | 0 | $\sim 6 s$ |
> > > |  |  |  | Poisson Equation |  |  |  |
> > > | BSL(Ours) | $1.18 \times 10^{-2}$ | $0.133$ | $16.7$ | 1 | 1 | $\sim 92(+9) s$ |
> > > | PINN-SR | Fail | Fail | Fail | 0 | 0 | $\sim 3737 s$ |
> > > | SINDy | $1.15$ | 87 | 962 | 1 | 1 | $\sim 10 s$ |
> > > | RVM | $1.15$ | 232 | 968 | 1 | 1 | $\sim 10 s$ |
> > > ||||||||

---

> ### Author Response · Authors · 2022-08-08
> **Thank you for your insightful comments**
>
> Dear reviewer R79g,
>
> Again, we would like to thank your insightful comments on our paper. We believe we have fully addressed your concerns through our response earlier and the revised paper. We do hope to have your feedback and look forward to answering any additional questions you have. Thank you very much.

---

> ### Author Response · Authors · 2022-08-10
> **Friendly reminder**
>
> Dear Reviewer R79g,
>
> This is a friendly reminder that the discussion period is coming close to the end. Your comments and suggestions have been vital for helping improve the quality of our paper, which are greatly appreciated. We believe that your concerns have been fully addressed through our response below. We sincerely hope to have your feedback as well as your possible consideration for raising the rating score of our paper. Thank you very much.
>
> Best regards,
>
> The authors

---

### Author Response · Authors · 2022-08-04
**Revised manuscript uploaded and response to reviewers' comments posted**

Dear reviewers,

We sincerely thank you for your constructive comments and suggestions, which helps to improve our manuscript. We have posted the response, the revised version of the paper and the supplementary materials (changes marked in blue). We include more evaluation metrics and benchmark ODE/PDE cases, including one real-world data set. Moreover, we have added more discussions and experiment details accordingly. We are looking forward to your further feedback. Please do feel free to let us know if you have any thoughts.

Thank you for your efforts.

Best regards,

The authors

---

### Meta-Review · Area_Chair_syVP · 2022-08-21

**Recommendation:** Accept
**Confidence:** Less certain

**Metareview:**

The reviewers are all leaning towards acceptance, with two having acknowledged the extensive author feedback. I encourage the authors to incorporate as clearly as possible the extensive work done during the rebuttal period into the text when updating their paper for publication.

**Award:**

No

---

### Decision · Program_Chairs · 2022-09-14

Accept